# LiveCodeBench: Holistic and Contamination Free Evaluation of Large Language Models for Code

**Naman Jain**[†]   **King Han**[†]   **Alex Gu**[* $]   **Wen-Ding Li**[*‡]   **Fanjia Yan**[*†]   **Tianjun Zhang**[*†]

**Sida I. Wang**   **Armando Solar-Lezama**[$]   **Koushik Sen**[†]   **Ion Stoica**[†]

[†] UC Berkeley       [$] MIT       [‡] Cornell

{naman_jain,kingh0730,fanjiayan,tianjunz,ksen,istoica}@berkeley.edu
{gua,asolar}@csail.mit.edu   wl678@cornell.edu   sidawang88@gmail.com

## Abstract

Large Language Models (LLMs) applied to code-related applications have emerged as a prominent field, attracting significant interest from academia and industry. However, as new and improved LLMs are developed, existing evaluation benchmarks (e.g., HumanEval, MBPP) are no longer sufficient for assessing their capabilities suffering from data contamination, overfitting, saturation, and focus on merely code generation. In this work, we propose LiveCodeBench, a comprehensive and contamination-free evaluation of LLMs for code, which collects *new* problems over time from contests across three competition platforms, LeetCode, AtCoder, and CodeForces. Notably, our benchmark also focuses on a broader range of code-related capabilities, such as self-repair, code execution, and test output prediction, beyond just code generation. Currently, LiveCodeBench hosts over six hundred coding problems that were published between May 2023 and Aug 2024. We evaluate over 50 LLMs on LiveCodeBench (LCB for brevity) presenting the largest evaluation study of code LLMs on competition problems. Based on the study, we present novel empirical findings on contamination, overfitting, and holistic evaluations. We demonstrate that time-segmented evaluations serve as a robust approach to evade contamination; they are successful at detecting contamination across a wide range of open and closed models including GPT-4-O, Claude, DeepSeek, and Codestral. Next, we highlight overfitting and saturation of traditional coding benchmarks like HumanEval and demonstrate LCB allows more reliable evaluations. Finally, our holistic evaluation scenarios allow for measuring the different capabilities of programming agents in isolation.

## 1 Introduction

Code has emerged as an important application area for LLMs, with a proliferation of code-specific models (Chen et al., 2021; Austin et al., 2021; Nijkamp et al., 2022; Li et al., 2023b; Roziere et al., 2023; Guo et al., 2024; Ridnik et al., 2024; Lozhkov et al., 2024) and their applications across various tasks such as program repair (Zheng et al., 2024; Olausson et al., 2023), optimization (Madaan et al., 2023a), test generation (Steenhoek et al., 2023), documentation (Luo et al., 2024), SQL (Sun et al., 2023). In contrast with these rapid advancements, evaluations have remained relatively stagnant, and current benchmarks like HumanEval and MBPP paint a misleading picture. Firstly, while coding is a multi-faceted skill, these benchmarks only focus on code generation, thus overlooking broader code-related capabilities. Moreover, these benchmarks suffer from contamination or overfitting, as benchmark samples are present in the training datasets.

---

[1]Note that for model comparisons – performances are averaged across multiple months and platforms achieving a larger sample size.

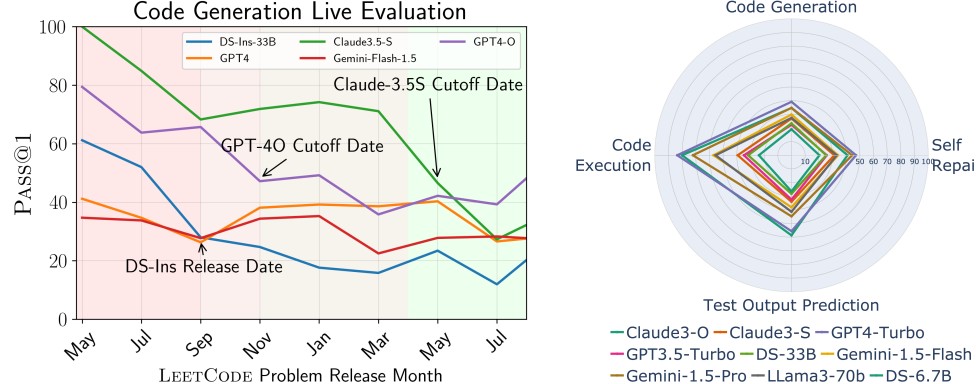

Figure 1: **Left.** LIVECODEBENCH comprises problems marked with release dates, allowing evaluations over different time windows. The figure depicts performances of DEEPSEEK, GPT-4-O, and CLAUDE-3S models over bimonthly time windows ($\sim 40$ LEETCODE problems) showcasing a stark drop after their cutoff dates, highlighting contamination. Thus, we can detect and avoid contamination by evaluating models only on time-windows after the model's cutoff date (green region)[1]. **Right.** We evaluate LLMs across four scenarios that capture key coding capabilities necessary for building programming agents: code generation, repair, testing, and comprehension. The figure depicts various model performances across the scenarios available in LIVECODEBENCH in a radial plot – highlighting relative differences changing across the scenarios.

Motivated by these shortcomings, we introduce LIVECODEBENCH, built on the following principles:

1. **Live updates to prevent contamination.** LLMs are trained on massive inscrutable corpora and current benchmarks suffer from the risk of data contamination as they could be included in those training datasets. While previous works have attempted decontamination using both exact and fuzzy matches (Li et al., 2023b;d), it can be a non-trivial task (Team, 2024; Yang et al., 2023). Existing competition programming benchmarks (like APPS and CODESCOPE) are already contaminated and may fail to provide reliable evaluation of code LLM capabilities. In LIVECODEBENCH, to prevent the risk of problem contamination, we evaluate models on *new* problems tagged with a *release date*. Next, for newer models, we only consider problems released after the model's cutoff date to ensure that the model has not been trained on it as demonstrated in Figure 1 left.

2. **Holistic Evaluation.** Current code evaluations primarily focus on natural language to code generation. However, programming is a multi-faceted task that requires capabilities beyond those measured by code generation. These broader capabilities are even more relevant for constructing programming agents that can interact with the execution environment. Therefore, we evaluate the execution-feedback-based multi-turn coding using the self-repair scenario (Olausson et al., 2023), assess code comprehension capabilities using the code execution scenario (Gu et al., 2024), and introduce the test output prediction scenario to evaluate the models' test generation capabilities.

3. **High-quality problems and tests.** High-quality problems and tests are crucial for reliable evaluation of LLMs. However, existing benchmarks suffer from multiple deficiencies. First, existing competitive programming benchmarks (APPS, CODE-CONTESTS, xCODEEVAL, CODESCOPE) contain problems not amenable to input-output-based auto-grading. For example, CODE-CONTESTS (Li et al., 2022) (page 39, second-to-last paragraph) reports that about twenty-five percent of the problems in the benchmark accept multiple correct outputs for a single input. This *incorrectly penalizes correct solutions*, adding noise to a considerable fraction of the benchmark. Next, prior benchmarks like HUMANEVAL and APPS contain insufficient tests, further exacerbating noise in evaluations ( Liu et al. (2023a) reports $8\%$ drop in model performance). In LIVECODEBENCH, we source high-quality problems from reputable platforms and implement heuristics to detect and remove problems not amenable to input-output-based auto-grading. Finally, for every problem, we provide a substantial number of tests (over $18$ on average) for reliable and efficient evaluations. In contrast to

Table 1: Comparing LCB with existing coding and competition programming benchmarks.

| Benchmark | Contamination Prevention | Problem Curation | Robust Test Based Eval. | Varied Difficulties | Not Saturated | Broader Eval. Scenarios | Comp. Analysis across Models |
|---|---|---|---|---|---|---|---|
| HUMANEVAL (Chen et al., 2021) | ✗ | ✓ | ✗ | ✓ | ✗ | ✗ | ✓ |
| HUMANEVAL+ (Liu et al., 2023b) | ✗ | ✓ | ✓ | ✓ | ✗ | ✗ | ✓ |
| MBPP (Austin et al., 2021) | ✗ | ✓ | ✗ | ✓ | ✗ | ✗ | ✓ |
| APPS (Hendrycks et al., 2021) | ✗ | ✗ | ✗ | ✓ | ✓ | ✗ | ✗ |
| CODE-CONTESTS (Li et al., 2022) | ✗ | ✗ | ✓ | ✗ | ✓ | ✗ | ✗ |
| xCODEEVAL (Khan et al., 2023) | ✗ | ✗ | - | ✗ | ✓ | ✓[2] | ✗ |
| CODESCOPE (Yan et al., 2023) | ✗ | ✗ | - | ✗ | ✓ | ✓[2] | ✗ |
| TACO (Li et al., 2023c) | ✗ | ✗ | ✗ | ✗ | ✓ | ✗ | ✗ |
| USCAOBENCH (Shi et al., 2024) | ✗ | ✓ | ✓ | ✓ | ✓ | ✗ | ✗ |
| LIVECODEBENCH (Ours) | ✓ | ✓ | ✓ | ✓ | ✓ | ✓ | ✓ |

prior works, we also include several large tests designed for stress-testing solutions ensuring weak or worse complexity solutions do not pass test harnesses.

4. **Difficulty Guided Problem Curation.** Competitive programming is a challenging domain even for strong LLMs. As a result, these problems can be unsuitable for meaningfully comparing today's open LLMs, because the variance in performance is low, often relying on less than 1% performance differences (within the margin of error). Therefore, we use problem difficulty ratings (sourced from the competition websites) to curate our problems, avoiding those that are too difficult for current models. In particular, we avoid collecting CODE-FORCES problems used by prior works (CODE-CONTESTS, CODESCOPE, xCODEEVAL) since we find they do not sufficiently distinguish models due to model performances (PASS@1) tending to zero. Indeed, LCB easy and medium problems can separate 7B models that are indistinguishable on the hard subset (see DS-7B vs SC2-7B).

With these principles in mind, we build LIVECODEBENCH, a continuously updated benchmark that avoids data contamination. Particularly, we collect 612 problems from contests occurring between May 2023 and Aug 2024 and use them to construct the different scenarios.

**Empirical Findings.** We have evaluated over 50 (18 base models and 34 instruction-tuned) models across different LIVECODEBENCH scenarios. Based on this study and following analysis, we present novel empirical findings which have not been revealed in prior benchmarks.

1. **Contamination.** We observe a stark drop in the performance of DEEPSEEK, GPT-4-O, CODESTRAL and CLAUDE-3S on LEETCODE problems released after Aug 2023, Oct 2023, Jan 2024, and April 2024 (Figure 1 left). These results highlight likely contamination in older problems and time-segmented evaluations prove effective for performing fair comparisons across varied set of models from both open and closed domains.

2. **Holistic Evaluation.** Our evaluations reveal that model performances are correlated across tasks, but the relative differences do vary. For example, in Figure 1 right, the gap between open and closed models further increases on tasks like self-repair or test output prediction. Similarly, CLAUDE-3-OPUS surpasses GPT-4 on the test output prediction scenario.

3. **HUMANEVAL Overfitting.** Upon comparing LIVECODEBENCH with HUMANEVAL, we find that models cluster into two groups, ones that perform well on both benchmarks, and others that perform well on HUMANEVAL but not on LIVECODEBENCH (see Figure 4). The latter group primarily comprises fine-tuned open-access models while the former comprises base and closed models. This indicates some level of overfitting to HUMANEVAL.

4. **Model Comparisons** (Figure 3). We provide model comparisons across different groups like base, instruct, open, and closed models and across groups like open vs closed models.

**Concurrent Work.** (Huang et al., 2023) also evaluate LLMs in a time-segmented manner. However, they only focus on CODEFORCES problems, while we combine problems across platforms and additionally propose a holistic evaluation across multiple code-related scenarios. Liu et al. (2024) evaluate the code comprehension capabilities of LLMs using execution. (Singhal et al., 2024) also evaluates LLMs on more tasks but focus on *non-functional-correctness aspects* of programming. Shi et al. (2024) evaluate LLMs on USACO problems while we focus on simple competition scenarios.

---

[2]LIVECODEBENCH considers different evaluation scenarios compared to xCODEEVAL and CODESCOPE focusing on settings where reliable evaluation is possible.

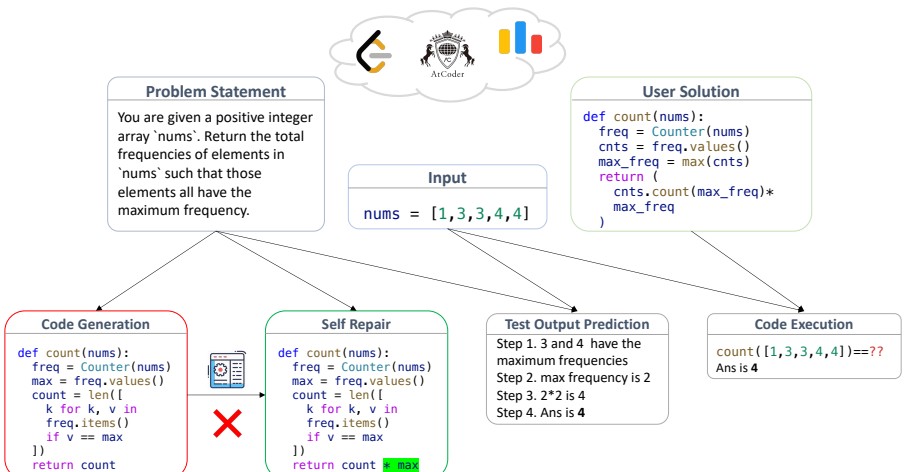

Figure 2: Overview of the four scenarios present in LIVECODEBENCH.

## 2   HOLISTIC EVALUATION

Code capabilities of LLMs are evaluated and compared using natural language to code generation tasks. However, this only captures one dimension of code-related capabilities. AlphaCodium (Ridnik et al., 2024) developed an intricate LLM pipeline for solving competition coding problems. By combining natural language reasoning, test case generation, code generation, and self-repair, they achieve significant improvements over a naive direct code generation baseline, showcasing the importance of these broader capabilities. Motivated by this, we propose a holistic evaluation of LLMs using a suite of evaluation setups that capture a broader range of code-related capabilities.

Specifically, we evaluate code LLMs in four scenarios, namely code generation, self-repair, code execution, and test output prediction. Our selection criterion was to pick settings that are useful components in code LLM workflows and in addition, have clear and automated evaluation metrics.

Following we describe each of these scenarios in detail.

**Code Generation.** The code generation scenario follows the standard setup for generating code from natural language. The model is given a problem statement, which includes a natural language description and example tests (input-output pairs), and is tasked with generating a correct solution. The evaluation is performed based on functional correctness, using a set of *unseen* test cases. We use the PASS@1 metric measured as the fraction of the problems for which the model was able to generate a program passing all tests. Figure 2 (left) provides an example of this scenario.

**Self Repair.** The self-repair scenario is based on previous works that tested the self-repair capabilities of LLMs (Olausson et al., 2023; Shinn et al., 2023; Chen et al., 2023). Here, the model is given a problem statement from which it generates a candidate program (similar to the single-step code generation scenario above). However, in case of a mistake, the model is additionally provided with error feedback (either the exception message or a failing test case) and is tasked with generating the fixed solution. Similar to the code generation scenario, the evaluation is performed via functional correctness on the final program, i.e. either the single-step correct generation or the attempted repair. We use the PASS@1 metric to measure the combined performance after the repair step. Figure 2 (mid-left) provides an example of this scenario.

**Code Execution.** The code execution scenario is based on the output prediction setup used in CRUXEVAL (Gu et al., 2024). The model is provided a program snippet consisting of a function (f) along with a test input to the program and is tasked with predicting the output of the program on the input test case. The evaluation is performed via an execution based on an exact match correctness metric. Figure 2 (right) provides an example of the code execution scenario.

**Test Case Output Prediction.** Finally, we introduce a new task that is designed to study natural language reasoning and test generation. In this task, the model is given the problem statement along with a test case input, and it is tasked with generating the expected output for that input. This task follows a setup similar to the one used in CODET (Chen et al., 2022), where tests are generated solely

from problem statements, without the need for the function's implementation. A key difference is that we provide a fixed set of test inputs for each problem in our dataset, and the models are then prompted to only predict the expected output for those specific inputs. This approach allows for a straightforward evaluation of the test generation capabilities by avoiding test input prediction, a hard-to-evaluate task. Figure 2 (mid-right) provides an example of this scenario.

## 3 BENCHMARK CURATION

We curate our problems from three coding competition websites: LEETCODE, ATCODER, and CODE-FORCES. These websites periodically host contests containing problems that assess the coding and problem-solving skills of participants. The problems consist of a natural language problem statement along with example input-output examples, and the goal is to write a program that passes a set of hidden tests. Further, thousands of participants participate, solving these problems thus ensuring that the problems are vetted for clarity.

### 3.1 DATA COLLECTION

We have written automated HTML scrapers for each of the above websites to collect problems and the corresponding metadata. To ensure quality and consistency, we parse mathematical formulas and exclude problems with images. We also exclude problems that are not suitable for grading by input-output examples, such as those that accept multiple correct answers or require the construction of data structures. Specifically, we use keyword-based heuristic filters to filter interactive problems and problems accepting multiple correct solutions. Besides parsing the problem descriptions, we also collect associated ground truth solutions and test cases whenever directly available. Thus for each problem, we collect tuples of natural language problem statement $P$, test cases $T$, and ground truth solution $S$. Finally, we associate the contest date $D$ to mark the release date of each problem and use the collected attributes to construct problems for our four scenarios (detailed in Section C.2 ahead). Note that this process is completely automated and human involvement is only involved in modifying high-level design decisions such as updating problem difficulty settings and improving the keyword-based heuristics for filtering problems over different "live updates".

**Scrolling through time.** As noted, we associate the contest date $D$ for each problem. The release date allows us the measure the performance of LLMs over different time windows by filtering problems based on whether the problem release date falls within a time window (referred to as "scrolling" through time). This is crucial for evaluating and comparing models trained at different times. Specifically, for a new model and the corresponding cutoff date (normalized to the release date if the training cutoff date is not published), we can measure the performance of the model on benchmark problems released after the cutoff date. We have developed a UI that allows comparing models on problems released during different time windows (Figure 10).

**Test collection.** Tests are crucial for assessing the correctness of the generated outputs and are used in all four scenarios. We collect tests available on platform websites whenever possible and use them for the benchmark. Otherwise, following (Liu et al., 2023b), we use a LLM (here GPT-4-TURBO) to generate test *inputs* for the problems. A **key difference** between our test generation approach is that instead of generating inputs directly using the LLM, we construct generators that sample inputs based on the problem specifications using in-context learning. Details of this approach can be found in Section A.2. Finally, we collect a small fraction of failing tests from the platforms for robust adversarial tests. Section A.4 describes our design decisions to determine the number of tests.

**Problem difficulty.** Competition programming has remained a challenge for LLMs, with GPT-4 achieving an average CODEFORCES rating (ELO) of 392, placing it in the bottom 5 percentile (OpenAI, 2023). This makes it difficult to compare LLMs, as the variation in performance across models is low. In LIVECODEBENCH, we collect problems of diverse difficulties as labeled in competition platforms, excluding problems that are rated above a certain threshold that are likely too difficult for even the best models[3]. Further, we use these ratings to classify problems as EASY, MEDIUM, and HARD for more granular model comparisons, estimating difficulty roughly based on HUMANEVAL scores.

We defer the platform and scenario specific curation details to the Appendix (Section C)

---

[3]From our early explorations, we find CODEFORCES problems being considerably more difficult than ATCODER and LEETCODE problems and thus focus primarily on the latter platforms.

## 4 EXPERIMENT SETUP

We describe the experimental setup in this section. First, we provide the common setup across the scenarios, followed by the scenario-specific setups in Section 4.1.

**Models.** We evaluate 52 models across various sizes, ranging from 1.3B to 70B, including base models, instruction models, and both open and closed models. Our experiments include models from different classes, such as GPTs, CLAUDES, GEMINIS, MISTRAL, LLAMA-3S, DEEPSEEKS, CODELLA-MAS, STARCODER2, CODEQWEN and their variants. Appendix D.1 provides the list of models.

**Evaluation Metrics.** We use the PASS@1 (Kulal et al., 2019; Chen et al., 2021) metric for our evaluations. Specifically, we generate 10 candidate answers for each problem from model providers or using VLLM (Kwon et al., 2023). We use nucleus sampling with temperature 0.2 and top_p 0.95 and calculate the fraction of correct solutions.

### 4.1 SCENARIO-SPECIFIC SETUP

The setup for each scenario is presented below. Note that the base models are only used in the code generation scenario since they do not easily follow the format for the other scenarios.

**Code Generation.** For the instruction-tuned models, we use a zero-shot prompt and follow the approach of Hendrycks et al. (2021) by adding appropriate instructions to generate solutions in either functional or stdin format (one-shot for base-models). Section D.2 depicts the prompt.

**Self Repair.** Similar to prior work Olausson et al. (2023), we use the programs generated during the code generation scenario along with the corresponding error feedback to build the zero-shot prompt for the self-repair scenario. The error feedback can be syntax errors, runtime errors, wrong answers, and time-limit errors. Section D.3 provides error feedback and the corresponding prompt.

**Code Execution.** We use few-shot prompts for the code execution scenario, both with and without chain-of-thought prompting (COT). Particularly, we use a 2-shot prompt without COT and a 1-shot prompt with COT with manually detailed steps. The prompts are detailed in Section D.4.

**Test Output Prediction.** We use a zero-shot prompt that queries the model to complete assertions, given the problem, function signature, and test input. We provide the prompt in Section D.5.

## 5 RESULTS

We first describe how LIVECODEBENCH helps detect and avoid benchmark contamination in Section 5.1. Next, we present the findings from our evaluations on LIVECODEBENCH in Section 5.2.

### 5.1 AVOIDING CONTAMINATION

**Contamination in DEEPSEEK and GPT-4-O.** LIVECODEBENCH curates problems released since May 2023. However, DEEPSEEK was released Sep 2023 and GPT-4-O's official cutoff date is Nov 2023. We can measure the performance of these models on the benchmark problems released after these dates, thereby estimating the performance of the model on previously unseen problems. Figure 1 shows the performance of these models on LEETCODE problems released in different months from May 2023 and Aug 2024. First, we notice stark drops in model performance – DS-INS-33B model after Aug. 2023, GPT-4-O model after Nov. 2023, CLAUDE-3S model after Apr. 2023. These drops align with their release or cutoff dates which suggests that the earlier problems might indeed be contaminated. This trend is consistent across other LIVECODEBENCH scenarios like repair and code execution, as depicted in Figure 11. Concurrently, (Guo et al., 2024) (Section 4.1, last paragraph) also acknowledge the possibility of LEETCODE contamination, noting that "*models achieved higher scores in the LeetCode Contest held in July and August*".

Interestingly, we find that this drop in performance primarily occurs for the LEETCODE problems only and that the model performance is relatively smooth across the months for problems from other platforms. Figure 12 shows a relatively stable performance for all models on ATCODER problems.

**Performances of other models.** We study performance variations in other models released more recently. Particularly, GPT-4-TURBO, MISTRAL-L, and CLAUDE-3S models were released in Nov'2023,

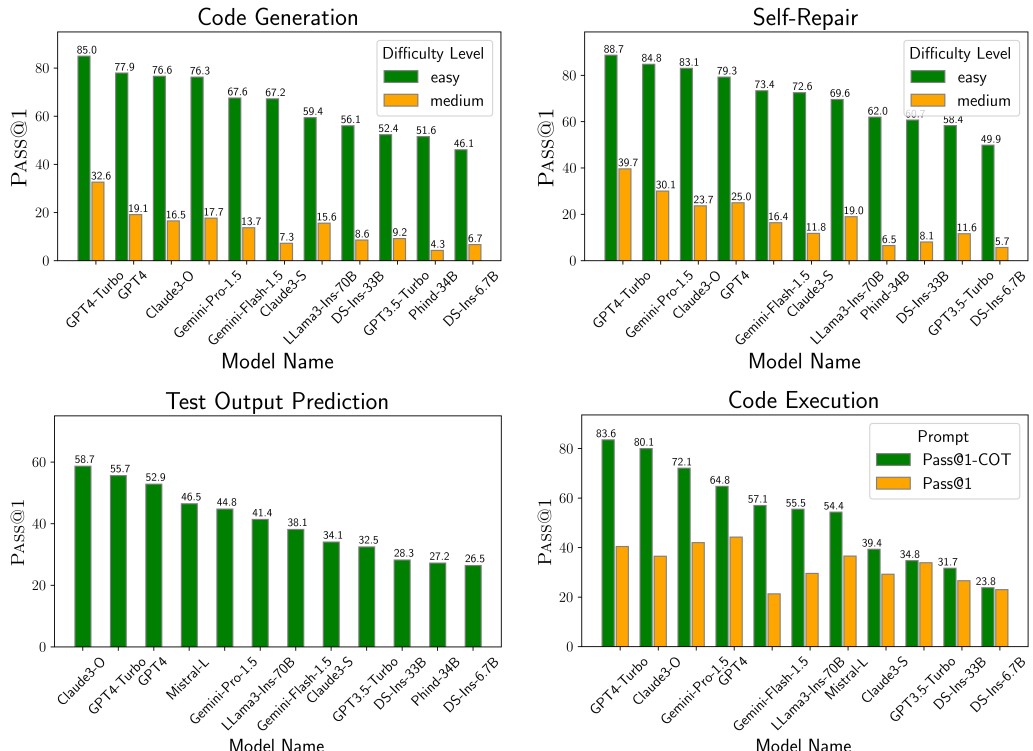

Figure 3: Model performances across the four scenarios available in LIVECODEBENCH .

Feb'2024, and Mar'2024 respectively. Empirically, we do not observe significant performance variations across the months, as shown in Figure 13.Interestingly, we find that even the DS-BASE-33B model also suffers from contamination dropping from PASS@1 $\sim$ 60 in May problems to PASS@1 $\sim$ 0 in September LEETCODE problems. Finally, CODESTRAL achieves PASS@1 36.5 on problems released between May'23 and Jan'24 and PASS@1 28.3 on problems post Jan'24.

## 5.2 PERFORMANCE AND MODEL COMPARISONS

We provide our model comparison findings here (and in Appendix E). To overcome contamination issues in DEEPSEEK models, we only consider problems released since Sep 2023 for all evaluations below. Figure 3 shows the performance of a subset of models across the four scenarios.

**Holistic Evaluations.** We have evaluated the models across the four scenarios currently available in LIVECODEBENCH. Figure 1 displays the performance of models on all scenarios along the axes of the polar chart. First, we observe that the relative order of models remains mostly consistent across the scenarios. This is also supported by high correlations between PASS@1 metric across the scenarios – over 0.88 across all pairs as shown in Figure 14. However, despite the strong correlation, the relative differences in performance do vary across the scenarios. For example, GPT-4-TURBO further gains performance gap over GPT-4 in the self-repair scenario after already leading in the code generation scenario. Similarly, CLAUDE-3-OPUS and MISTRAL-L perform well in tasks involving COT, particularly in the code execution and test output prediction scenarios. For instance, CLAUDE-3-OPUS even outperforms GPT-4-TURBO in the test output prediction scenario. These differences highlight the need for holistic evaluations beyond measuring code generation capabilities.

**Comparison to HUMANEVAL.** Next, we compare how code generation performance metrics translate between LIVECODEBENCH and HUMANEVAL, the primary benchmark used for evaluating coding capabilities. Note that we use HUMANEVAL+ providing more accurate evaluations. Figure 4 shows a scatter plot of PASS@1 on HUMANEVAL+ versus LCB-Easy code generation scenario. We find only a moderate correlation of 0.72, with much larger performance variations on LCB-Easy.

Additionally, we observe that the models cluster into two groups, shaded in red and green. The models in the green-shaded region lie close to the $x = y$ line, indicating that they perform similarly on both

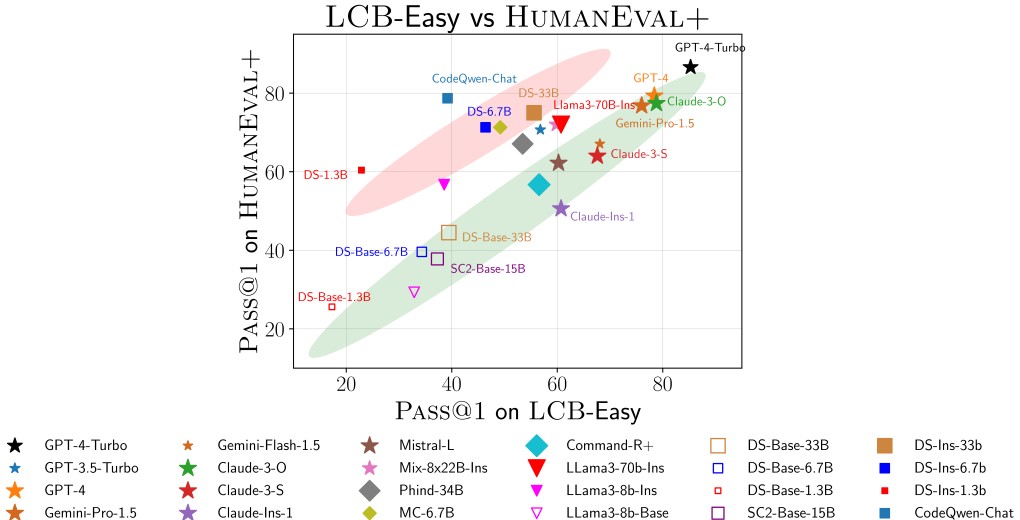

Figure 4: Scatter plot comparing PASS@1 of models on HUMANEVAL+ versus LCB-Easy (time-window Sep 2023 to May 2024). We find that the models are separated into two groups – the green-shaded region where performances on the two datasets are *aligned* and the red-shaded region where models perform well on HUMANEVAL+ but perform poorly on LIVECODEBENCH. This indicates potential overfitting on HUMANEVAL+ and primarily occurs in the fine-tuned variants of open-access models. For example, DS-INS-1.3B achieves PASS@1 of 60 and 26 on HUMANEVAL+ and LCB-Easy subset. Thus, while it ranks above CMD-R+ (∼ 100x larger!) on HUMANEVAL+, it performs significantly worse on the LCB. Similarly, DS-INS-6.7B and CODEQWEN outperform SONNET on HUMANEVAL+ but are > 20 points behind on LCB-Easy.

benchmarks. On the other hand, models in the red shaded region perform well on HUMANEVAL+ but not as well on LIVECODEBENCH. Interestingly, the green-shaded cluster contains base models or closed-access models, while the red-shaded cluster primarily comprises fine-tuned variants of open-access models. The well-separated clusters suggest that many models that perform well on HUMANEVAL might be overfitting on the benchmark, and their performances do not translate well to problems from other domains or difficulty levels like those present in LIVECODEBENCH. Indeed, HUMANEVAL has small and isolated programming problems and thus easier to overfit. In contrast, LIVECODEBENCH problems are sourced from reputable coding platforms offering more challenging problems with higher diversity and difficulty levels. We detail instances of this overfitting (DS-INS-1.3B, DS-INS-6.7B, and CODEQWEN) in Figure 4 caption.

**Comparing Base Models.** We use four families of base models and compare them on the code generation scenario. A one-shot prompt is used for all models to avoid any formatting and answer extraction issues. We find L3-BASE and DS-B models are significantly better than both CODELLAMA and STARCODER2 base models with a DS-BASE-6.7B model even outperforming both CL-BASE-34B and SC2-BASE-15B models. Note that some LCB specific differences can potentially be attributed to data curation approaches. For instance, SC models (and potentially DS as discussed in Section 5.1) use competition problems during pre-training.

**Role of Post Training.** Post-training improves performance on both HUMANEVAL+ and LIVE-CODEBENCH for the code generation scenario. Particularly, on LCB L3-INS-70B, DS-INS-33B and PHIND-34B improve PASS@1 over their base models by 8.2, 7.3 and 9.5 points respectively. This highlights the importance of good post-training datasets for building strong LLMs. At the same time, we note that the base models have *aligned* performances on LCB code generation and HUMANEVAL+ benchmarks and lie within or close to the green shaded region in Figure 4. However, the fine-tuned open models exhibit a larger performance gap, with much better performances on HUMANEVAL+. On the other hand, the closed-access models are still aligned across both benchmarks. This suggests the necessity of more diverse post-training data mixtures.

**Open-Access vs Closed-Access Models.** In general, closed (API) access model families generally outperform the open access models. The gap is only closed by three models, namely L3-INS-70B, MIXTRAL, and DS-INS-33B which reach the performance levels of the closed models. For instance, in

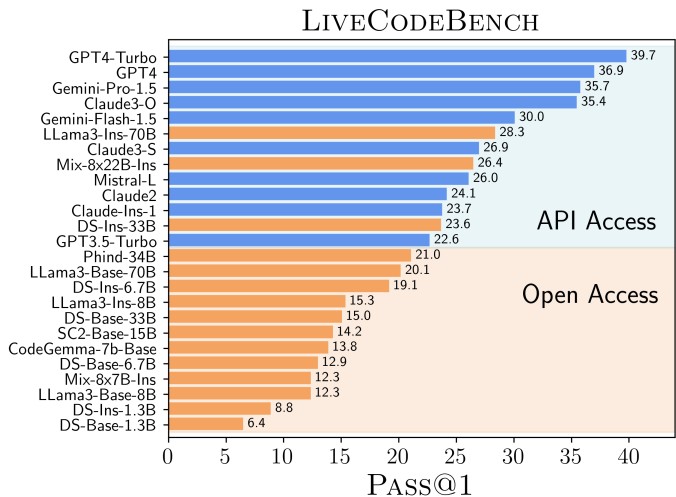

Figure 5: Comparison of open access and (closed) API access models on LIVECODEBENCH code generation scenario. We find that closed-access models consistently outperform the open models with only strong instruction-tuned variants of > 30B models (specifically L3-INS-70B, MIXTRAL and DS-INS-33B models) crossing the performance gap.

the code generation scenario (Figure 5), these models reach close to or even outperform closed access models like GEMINI-PRO, GPT-3.5-TURBO, and CLAUDE-3-SONNET. The performances vary across scenarios with the closed-access models faring better in test output and code execution scenarios.

**Highlighting gap between SoTA** One distinct observation from our evaluations is the large gap between SoTA models and open models across all scenarios. Particularly, GPT-4-TURBO, GPT-4, GEMINI-PRO-1.5 and CLAUDE-3-OPUS lead across the benchmarks with wide performance margins over other models. This distinguishes LIVECODEBENCH from prior benchmarks (like HUMANEVAL) where various open models have achieved similar or better performance. For example, DS-INS-33B is merely 4.3 point behind GPT-4-TURBO on HUMANEVAL+ but 16.2 points (69%) on LCB code generation scenario. This gap either holds or sometimes even amplifies across other scenarios.

**Comparing open-access instruction-tuned models.** Here, we compare various fine-tuned variants of the L3-BASE, DEEPSEEK and CODELLAMA base models across different model sizes. We find that fine-tuned L3-BASE and DEEPSEEK models lead in performance, followed by PHIND-34B and CODELLAMA models across most scenarios. Broadly, we find that model performances correlate with model sizes. For example, PHIND-34B model outperforms the 6.7B models across all scenarios.

# 6 RELATED WORK

**Language Models for Code Generation.** Starting with Codex (Chen et al., 2021), there are over a dozen code LLMS. These include CodeT5 (Wang et al., 2021; 2023), CodeGen (Nijkamp et al., 2022), SantaCoder (Allal et al., 2023), StarCoder (Li et al., 2023b; Lozhkov et al., 2024), InCoder (Fried et al., 2022), CodeGeeX (Zheng et al., 2023), L3-BASE, DEEPSEEK (Bi et al., 2024) and CODELLAMA (Roziere et al., 2023).

**Code Generation Benchmarks.** Many benchmarks have been proposed to compare and evaluate these models. These primarily focus on natural language to Python code generation: HUMANEVAL (Chen et al., 2021), HUMANEVAL+ (Liu et al., 2023b), APPS (Hendrycks et al., 2021), CODE-CONTESTS (Li et al., 2022), MBPP (Austin et al., 2021), L2CEval (Ni et al., 2023). Their variants have been proposed to cover more languages, (Wang et al., 2022a; Zheng et al., 2023; Cassano et al., 2022; Athiwaratkun et al., 2022). Many benchmarks have focused on code generation in APIs. Benchmarks like DS-1000 (Lai et al., 2023), ARCADE (Yin et al., 2022), NumpyEval (Zhang et al., 2023b), and PandasEval (Jain et al., 2022) focus on data science APIs. Other benchmarks measure using broader APIs or general software engineering tasks, such as JuICe (Agashe et al., 2019), APIBench (Patil et al., 2023), RepoBench (Liu et al., 2023c), ODEX (Wang et al., 2022b), SWE-Bench (Jimenez et al.,

2023), GoogleCodeRepo (Shrivastava et al., 2023), RepoEval (Zhang et al., 2023a), ClasEval (Du et al., 2023) and Cocomic-Data (Ding et al., 2022).

A few benchmarks specifically measure competitive programming, such as APPS (Hendrycks et al., 2021), CodeContests (Li et al., 2022), CodeScope (Yan et al., 2023), xCodeEval (Khan et al., 2023), and LeetCode-Hard (Shinn et al., 2023), and TACO (Li et al., 2023c). Table 1 highlights differences between them. Methods such as AlphaCode (Li et al., 2022), AlphaCode 2 (Gemini Team et al., 2023), ALGO (Zhang et al., 2023d), Parsel (Zelikman et al., 2022), code cleaning (Jain et al., 2023), code explanations (Li et al., 2023a), analogical reasoning (Yasunaga et al., 2023), and AlphaCodium (Ridnik et al., 2024) have focused on improving LLMs.

## 6.1 Holistic Tasks and Contamination

**Code Repair.** (Chen et al., 2023; Olausson et al., 2023; Madaan et al., 2023b; Peng et al., 2023; Zhang et al., 2023c) have investigated self-repair for existing code LLM benchmarks. These methods use error feedback for models to improve inspiring our code repair scenario.

**Code Execution.** Code execution was first studied in (Austin et al., 2021; Nye et al., 2021) Live-CodeBench's execution scenario is particularly inspired by CRUXEval (Gu et al., 2024), a recent benchmark measuring the reasoning and execution abilities of code LLMs. We differ from CRUXEval in that our benchmark is live, and our functions are more complex and human-produced.

**Test Generation.** Test generation using LLMs has been explored in (Yuan et al., 2023; Schäfer et al., 2024; Tufano et al., 2022; Watson et al., 2020). Here, we decouple the test inputs and outputs for fair evaluations. Finally, some works have additionally studied other tasks and scenarios like type prediction (Mir et al., 2022; Wei et al., 2023; Malik et al., 2019), code summarization (LeClair et al., 2019; Iyer et al., 2016; Barone & Sennrich, 2017; Hasan et al., 2021; Alon et al., 2018), code security (Liguori et al., 2022; Pearce et al., 2022; Tony et al., 2023), etc.

**Contamination.** Data contamination and test-case leakage have received considerable attention Oren et al. (2024); Golchin & Surdeanu (2023); Weller et al. (2023); Roberts et al. (2024) recently. (Sainz et al., 2023) demonstrated contamination by simply prompting the model to highlight its contamination. Some detection methods have also been built to avoid these cases (Shi et al., 2023; Zhou et al., 2023). For code, (Riddell et al., 2024) use edit distance and AST-similarity to detect contamination. (Ma et al., 2021; Li et al., 2024) have explore "live" benchmark in other domains.

## 7 Limitations and Conclusion

We describe key limitations here and provide a deeper discussion in Appendix G.

**Evaluation Noise.** We anticipate noise from benchmark size, prompts, and sampling. For benchmark size, while 612 problems is fairly larger than existing code benchmarks (e.g. 164 problems in HumanEval), this set reduces on "scrolling" over the more recent problems. Next, prompting can cause large variations in model performance and we do not tune prompts across models. For sampling, we use bootstrapped Pass@1 using 10 completions, which should limit this noise considerably.

**Problem Domain.** LiveCodeBench comprises competition programming problems which might not correlate with how LLMs are used in practice. Even then, our results are well correlated with findings from human evaluations like Chatbot-Arena Chiang et al. (2024) thus providing useful signal.

**Conclusion.** In this work, we propose LiveCodeBench, a new benchmark for evaluating LLMs for code. LiveCodeBench provides an extensible framework that will keep on updating with new problems. Our benchmark mitigates contamination issues in existing benchmarks by introducing live evaluations and emphasizing scenarios beyond code generation to account for the broader coding abilities of LLMs. Our evaluations reveal novel findings such as contamination detection, holistic evaluations, and potential overfitting on HumanEval. We hope LiveCodeBench with serve to advance understanding of current code LLMs and also guide future research through our findings.

## ACKNOWLEDGEMENT

This work was supported in part by NSF grants CCF:1900968, CCF:1908870 and by SKY Lab industrial sponsors and affiliates Astronomer, Google, IBM, Intel, Lacework, Microsoft, Mohamed Bin Zayed University of Artificial Intelligence, Nexla, Samsung SDS, Uber, and VMware. A. Gu is supported by the NSF Graduate Research Fellowship under Grant No. 2141064. A. Solar-Lezama is supported by the NSF and Intel Corporation through NSF Grant CCF:2217064. Any opinions, findings, conclusions, or recommendations in this paper are solely those of the authors and do not necessarily reflect the position of the sponsors.

Finally, we thank Manish Shetty, Wei-Lin Chiang, Jierui Li, Horace He, Federico Cassano, Pengcheng Yin, Kexun Zhang, Rishabh Singh, and Aman Madaan for helpful feedback at various stages of the work. We particularly thank Kexun Zhang for providing results for C++ and Nikhil Thakur for helping maintain the benchmark.

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

# A DATASET

## A.1 LEGAL COMPLIANCE AND LICENSE

Our benchmark does not comprise and personal identifiable information or offensive content.

Similar to (Hendrycks et al., 2021), we scrape only the problem statements, ground-truth solutions, and test cases from competition websites – LEETCODE, ATCODER, and CODEFORCES. Further, we primarily scrape publicly visible portions of websites, avoiding any data collection that might be pay-walled or require login or interaction with the website. Following, (Hendrycks et al., 2021) we abide by Fair Use §107: "the fair use of a copyrighted work, including such use by ... scholarship, or research, is not an infringement of copyright", where fair use is determined by "the purpose and character of the use, including whether such use is of a commercial nature or is for nonprofit educational purposes", "the amount and substantiality of the portion used in relation to the copyrighted work as a whole", and "the effect of the use upon the potential market for or value of the copyrighted work." Finally, we use the collected problems for academic purposes only and in addition, do not train on the collected problems.

## A.2 GENERATOR BASED TEST GENERATION

We use GPT-4-TURBO to construct input generators. The following prompts (Figures 6 and 7) provide one-shot prompt templates used for synthesizing random and adversarial input generators. These generators define a function returns the arguments sampled in some distribution. These generators are then executed to construct inputs which validated on the collected correct programs. We use separate generators for random and adversarial setting since often times programming problems have corner cases which might not be captured by randomly sampling over the inputs. We build 2 random input generators, 4 adversarial input generators and check if the sampled inputs work for the correct programs. Finally, the number of collected inputs is thresholded to 100 for efficient grading (using random selection). We find that our generators can already function well but future work can study the design space of constructing such generators.

Note that for CODEFORCES, we construct the generators in semi-autonomous manner since only 9 problems were used.

## A.3 CODE EXECUTION

The code execution split of LiveCodeBench consists of 479 samples from 85 distinct problems. To encourage diversity in our benchmark while keeping our benchmark small and usable, we place a limit of six samples for each given problem. These sample programs and corresponding test cases are chosen uniformly at random from all those passing the filter.

**Filtering Criteria**: The specific filtering criteria are as follows:

- Compile time: length of code is between 100 and 500 characters, no syntax errors, all necessary imports are included
- Runtime: no floating point operations, true division, exp, other integer operations must have at least one argument $\leq 3$, string and list operations must have at least one argument with length $\leq 3$, finish running in 2 seconds, "reasonable" number of steps (roughly, under 1000 Python bytecode operations).

We give two examples of two programs that are filtered out in the Listings below. Our final benchmark consists of 479 samples from 85 problems, but will increase in size due to its live nature.

**Dataset Statistics**: As mentioned, we filter for codes between 100 and 500 characters, as well as below 1000 steps. The statistics for programs in our dataset are shown in Fig. 8.

## A.4 NUMBER OF TESTS

We can sample large amounts of diverse inputs using our generator-based input sampling approach. However, the number of inputs produces a tradeoff in terms of the usability of the benchmark, slowing

```
You are an expert python competitive programmar and your goal is to
    ↪ construct input-generators for testing programming contest
    ↪ problems. You will write relevant generators and finally
    ↪ construct `construct_inputs` function that returns a list of
    ↪ diverse inputs sampled from the generator. Remember to strictly
    ↪ follow the instructions and constraints present in the problem
    ↪ statement

<USER>
{EXAMPLE_PROBLEM}

Construct a random input generator.
</USER>

<ASSISTANT>
import numpy as np
def random_input_generator(weight_min, weight_max, size_min, size_max):
    weights_size = np.random.randint(size_min, size_max+1)
    weights = np.random.randint(weight_min, weight_max, size=
    ↪ weights_size).tolist()
    k = np.random.randint(1, len(weights)+1)
    return weights, k

def construct_inputs():
    inputs_list = []
    ## small inputs
    for i in range(15):
        inputs_list.append(random_input_generator(1, 10**3, 1, 10))
    ## medium inputs
    for i in range(15):
        inputs_list.append(random_input_generator(1, 10**6, 1, 10**3))
    ## large inputs
    for i in range(15):
        inputs_list.append(random_input_generator(1, 10**9, 1, 10**5))
    return inputs_list
</ASSISTANT>

<USER>
{PROBLEM}

Construct a random input generator. Use the format used in the above
    ↪ example by returning a single function that builds diverse
    ↪ inputs named `construct_inputs`
</USER>
```

Prompt for random input generation

Figure 6: Random Input Generator Prompt

down the evaluation process. Hence, we threshold the number of sampled inputs to less than 25 based on problem length, difficulty, and number of inputs.

We base this decision by estimating model performances when using different sets of inputs at random for testing in Figure 9. Particularly, we vary the number of private tests between 0, 1, 5, 10, 20, 40, 60, 80, and 100 and observe an exponentially decaying curve. We find the $\sim 20$ inputs provides a good threshold which balances evaluation time with evaluation rigor.

```
You are an expert python competitive programmar and your goal is to
    ↪ construct input-generators for testing programming contest
    ↪ problems. You will write relevant generators and finally
    ↪ construct `construct_inputs` function that returns a list of
    ↪ diverse inputs sampled from the generator. Remember to strictly
    ↪ follow the instructions and constraints present in the problem
    ↪ statement

<USER>
{EXAMPLE_PROBLEM}

Construct an adversarial input generator.
</USER>

<ASSISTANT>
import numpy as np
## case 1 - alternating large and small weights
def generate_adversarial_inputs_1(weight_size, max_weight, k):
    weights = [1 if i%2==0 else max_weight for i in range(weight_size)]
    return weights, k

## case 2 - equal_weights
def adversarial_input_generator_2(weight_size, max_weight, k):
    weights = [max_weight for _ in range(weight_size)]
    return weights, k

# Case 3 - Large weights at the ends
def adversarial_input_generator_3(weight_size, max_weight, k):
    pass ## truncatede
def construct_inputs():
    inputs_list = []

    weight_sizes = [10, 1000, 100000]
    max_weights = [10**3, 10**6, 10**9]

    for weight_size in weight_sizes:
        for max_weight in max_weights:
            ks = [1, 2, 5, weight_size//2, weight_size-1, weight_size]
            for k in ks:
                inputs_list.append(generate_adversarial_inputs_1(
    ↪ weight_size, max_weight, k))
                # truncated
    return inputs_list
</ASSISTANT>

<USER>
{PROBLEM}

Construct an adversarial input generator. Use the format used in the
    ↪ above example by returning a single function that builds diverse
    ↪  inputs named `construct_inputs`
</USER>
```

Prompt for adversarial input generation

Figure 7: Adversarial Input Generator Prompt

```python
def check(x, t):
    if x == '':
        return t == 0
    if t < 0:
        return False
    for i in range(len(x)):
        if check(x[:i], t - int(x[i:])):
            return True
    return False

@cache
def punishmentNumber(n: int) -> int:
    if n == 0:
        return 0
    ans = punishmentNumber(n-1)
    if check(str(n * n), n):
        ans += n * n
    return ans
assert punishmentNumber(n = 37) == 1478
```

Program filtered because of multiplication

```python
dp = [True for _ in range(int(1e6 + 5))]
MAXN = int(1e6 + 5)
p = []
dp[0] = False
dp[1] = False
for i in range(2, MAXN):
    if not dp[i]: continue
    p.append(i)
    for j in range(2 * i, MAXN, i):
        dp[j] = False
def findPrimePairs(n: int) -> List[List[int]]:
    res = []
    for i in range(1, n):
        if n % 2 == 1 and i > n//2: break
        if n % 2 == 0 and i > n//2: break
        if dp[i] and dp[n - i]:
            res.append([i, n - i])
    return res
assert findPrimePairs(n = 2) == []
```

Program filtered because of control flow

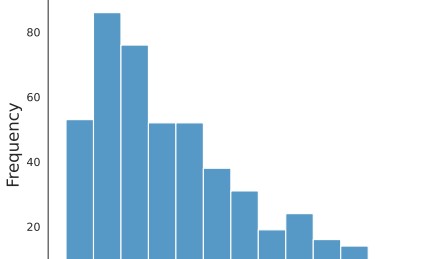 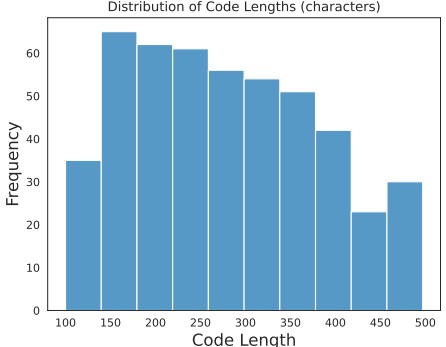

Figure 8: Distribution of code lengths and number of execution steps

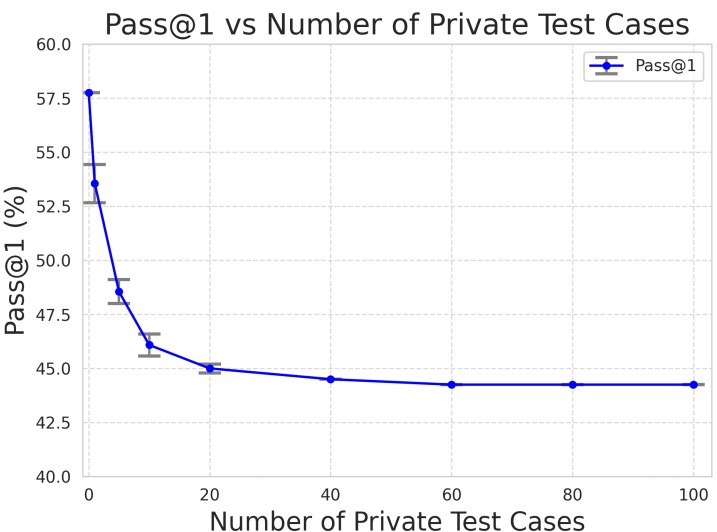

Figure 9: Model performance with varying counts of private inputs used for evaluation. We find the PASS@1 tends to stabilize when sampling about 20 inputs providing a good tradeoff in evaluation time and rigor.

# B  UI

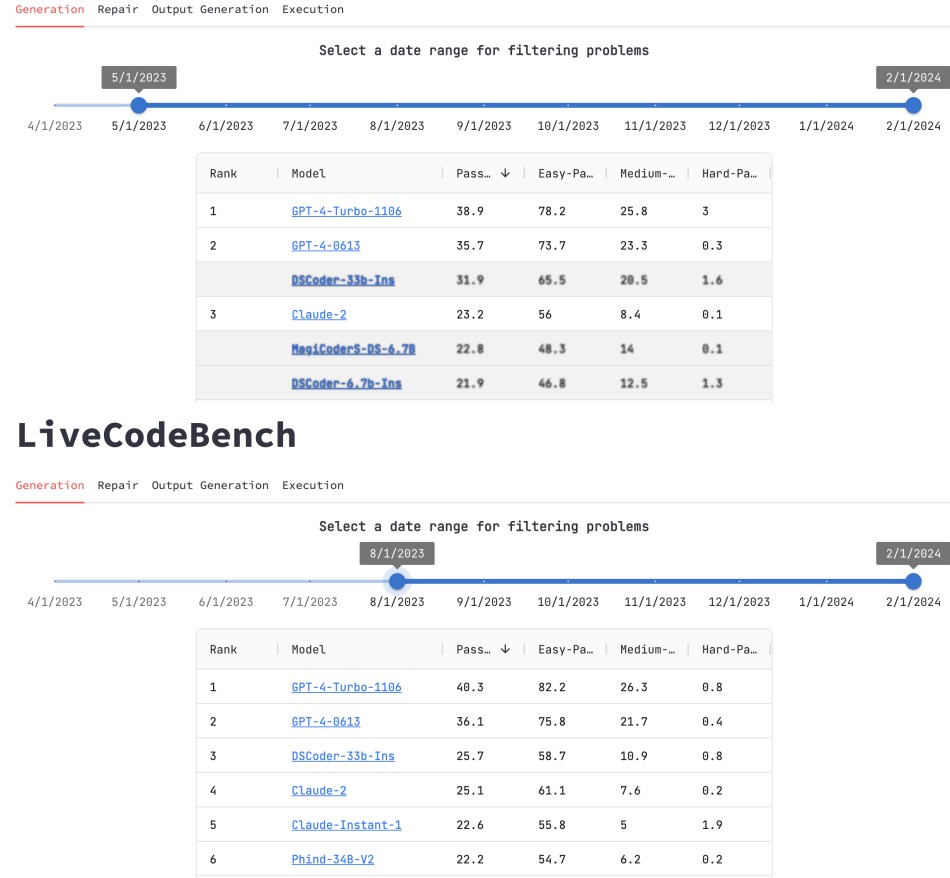

Figure 10: UI of LIVECODEBENCH showing two views – May-Jan and Sep-Jan. The contaminated models are blurred and the performance difference is visible across the two views. The scroller on the top allows selecting different periods of time highlighting the live nature of the benchmark.

## C CURATION

### C.1 PLATFORM SPECIFIC CURATION

We describe the curation process for each platform.

**LEETCODE.** We collect problems from all weekly and biweekly contests on LEETCODE that have taken place after April'23. For each problem, we collect the problems, public tests, and user solutions. The platform also provides a difficulty label for each problem which we use to tag the problems as EASY, MEDIUM, and HARD. Since LEETCODE provides a starter code for each problem, we also collect it and provide it to the LLM in the STDIN format. Since the hidden tests are not directly available, we use our generator-based test input generation approach (Section A.2) and also collect the auto grader failing tests for some of the recent problems.

**ATCODER.** We collect problems from the `abc` (beginner round) contests on ATCODER that have taken place after April'23. We deliberately avoid the more challenging `arc` and `agc` contests which are designed for more advanced Olympiad participants. The problems are assigned numeric difficulty ratings, and we exclude `abc` problems with a rating of more than $500$. We also use these numeric ratings to tag the problems as EASY, MEDIUM, and HARD. Specifically, we use the rating brackets $[0-200)$, $[200-400)$, and $[400-500]$ to perform the classification. ATCODER provides public and hidden tests for each problem which we directly use in the benchmark.

**CODEFORCES.** We have collected problems from the Division 3 and Division 4 contests on CODE-FORCES. Notably, we find that even with this filter, the problems are harder than the other two platforms. CODEFORCES also provides difficulty ratings for the problems which we use to tag the problems as EASY, MEDIUM, and HARD using the rating brackets $\{800\}$, $(800-1000]$, and $(1000-1300]$ respectively. Due to the higher difficulty, we only consider a small fraction of problems from CODE-FORCES and semi-automatically construct test case generators, as they do not provide complete tests on the platform (long tests are truncated).

Table 2 provides various statistics about the problems that we have collected for LIVECODEBENCH.

### C.2 SCENARIO-SPECIFIC BENCHMARK CONSTRUCTION

**Code Generation and Self-Repair.** We use the natural language problem statement as the problem statement for these scenarios. For LEETCODE, as noted above, an additional starter code is provided for the functional input format. For ATCODER and CODEFORCES problems, we use the standard input format (similar to (Hendrycks et al., 2021)). The collected or generated tests are then used to evaluate the correctness of the generated programs. Our final dataset consists of $511$ problem instances across the three platforms.

**Code Execution.** We draw inspiration from the benchmark creation procedure used in (Gu et al., 2024). First, we collect a large pool of $\sim 2000$ *correct, human-submitted solutions* from the

| Platform | Total Count | #Easy | #Medium | #Hard | Average Tests |
|---|---|---|---|---|---|
| LCB (May-end) | 511 | 182 | 206 | 123 | 17.0 |
| LCB (Sep-end) | 349 | 125 | 136 | 88 | 18.0 |
| ATCODER | 267 | 99 | 91 | 77 | 15.6 |
| LEETCODE | 235 | 79 | 113 | 43 | 19.0 |
| CODEFORCES | 9 | 4 | 2 | 3 | 11.1 |
| LCB-Easy | 182 | 182 | 0 | 0 | 16.1 |
| LCB-Medium | 206 | 0 | 206 | 0 | 17.4 |
| LCB-Hard | 123 | 0 | 0 | 123 | 18.0 |

Table 2: The statistics of problems collected in LIVECODEBENCH (LCB). We present the number of problems, their difficulty distributions and the average number of tests per problem. We present the results on the following subsets of LIVECODEBENCH (used throughout this manuscript) - (a) problems in the May-Feb and Sep-Feb time windows, (b) problems sourced from the three platforms, and (c) problems in the LCB-Easy, LCB-Medium, and LCB-Hard subsets.

LEETCODE subset. However, many of these programs have multiple nested loops, complex numerical computations, and a large number of execution steps. Therefore, we apply compile-time and run-time filters to ensure samples are reasonable, and we double-check this with a manual inspection. More details on the filtering criteria and statistics of the dataset can be found in Appendix A.3. Our final dataset consists of 479 samples from 85 problems.

**Test Case Output Prediction.** We use the natural language problem statement from the LEETCODE platform and the example test inputs to construct our test case output prediction dataset. Since the example test inputs in the problems are reasonable test cases for humans to reason about and understand the problems, they also serve as ideal test inputs for LLMs to process. Our final dataset consists of 442 problem instances from a total of 181 LEETCODE problems.

# D EXPERIMENTAL SETUP

## D.1 MODELS

We describe the details of models considered in our study in Table 3.

| Model ID | Short Name | Approximate Cutoff Date | Link |
|---|---|---|---|
| deepseek-ai/deepseek-coder-33b-instruct | DSCoder-33b-Ins | 08/30/2023 | deepseek-coder-33b-instruct |
| deepseek-ai/deepseek-coder-6.7b-instruct | DSCoder-6.7b-Ins | 08/30/2023 | deepseek-coder-6.7b-instruct |
| deepseek-ai/deepseek-coder-1.3b-instruct | DSCoder-1.3b-Ins | 08/30/2023 | deepseek-coder-1.3b-instruct |
| codellama/CodeLlama-70b-Instruct-hf | CodeLlama-70b-Ins | 01/01/2023 | CodeLlama-70b-Instruct-hf |
| openbmb/Eurus-70b-sft | Eurus-70B-SFT (n=1) | 01/01/2023 | Eurus-70b-sft |
| openbmb/Eurux-8x22b-nca | Eurux-8x22b-NCA (n=1) | 04/30/2023 | Eurux-8x22b-nca |
| codellama/CodeLlama-34b-Instruct-hf | CodeLlama-34b-Ins | 01/01/2023 | CodeLlama-34b-Instruct-hf |
| codellama/CodeLlama-13b-Instruct-hf | CodeLlama-13b-Ins | 01/01/2023 | CodeLlama-13b-Instruct-hf |
| codellama/CodeLlama-7b-Instruct-hf | CodeLlama-7b-Ins | 01/01/2023 | CodeLlama-7b-Instruct-hf |
| meta-llama/Meta-Llama-3-8B-Instruct | LLama3-8b-Ins | 01/01/2023 | Meta-Llama-3-8B-Instruct |
| meta-llama/Meta-Llama-3-70B-Instruct | LLama3-70b-Ins | 01/01/2023 | Meta-Llama-3-70B-Instruct |
| Phind/Phind-CodeLlama-34B-v2 | Phind-34B-V2 | 01/01/2023 | Phind-CodeLlama-34B-v2 |
| Smaug-2-72B | Smaug-2-72B | 01/01/2023 | Smaug-2-72B |
| Qwen-1.5-72B-Chat | Qwen-1.5-72B-Chat | 01/01/2023 | Qwen-1.5-72B-Chat |
| Qwen/CodeQwen1.5-7B | CodeQwen15-7B | 08/30/2023 | CodeQwen1.5-7B |
| Qwen/CodeQwen1.5-7B-Chat | CodeQwen15-7B-Chat | 08/30/2023 | CodeQwen1.5-7B-Chat |
| gpt-3.5-turbo-0301 | GPT-3.5-Turbo-0301 | 10/01/2021 | gpt-3.5-turbo-0301 |
| gpt-3.5-turbo-0125 | GPT-3.5-Turbo-0125 | 10/01/2021 | gpt-3.5-turbo-0125 |
| gpt-4-0613 | GPT-4-0613 | 10/01/2021 | gpt-4-0613 |
| gpt-4-1106-preview | GPT-4-Turbo-1106 | 04/30/2023 | gpt-4-1106-preview |
| gpt-4-turbo-2024-04-09 | GPT-4-Turbo-2024-04-09 | 04/30/2023 | gpt-4-turbo-2024-04-09 |
| gpt-4o-2024-05-13 | GPT-4O-2024-05-13 | 10/30/2023 | gpt-4o-2024-05-13 |
| claude-2 | Claude-2 | 12/31/2022 | claude-2 |
| claude-instant-1 | Claude-Instant-1 | 12/31/2022 | claude-instant-1 |
| claude-3-opus-20240229 | Claude-3-Opus | 04/30/2023 | claude-3-opus-20240229 |
| claude-3-sonnet-20240229 | Claude-3-Sonnet | 04/30/2023 | claude-3-sonnet-20240229 |
| claude-3-haiku-20240307 | Claude-3-Haiku | 04/30/2023 | claude-3-haiku-20240307 |
| codestral-latest | Codestral-Latest | 01/31/2024 | codestral-latest |
| gemini-pro | Gemini-Pro | 04/30/2023 | gemini-pro |
| gemini-1.5-pro-latest | Gemini-Pro-1.5-May | 04/30/2023 | gemini-1.5-pro-latest |
| gemini-1.5-flash-latest | Gemini-Flash-1.5-May | 04/30/2023 | gemini-1.5-flash-latest |
| ise-uiuc/Magicoder-S-DS-6.7B | MagiCoderS-DS-6.7B | 08/30/2023 | Magicoder-S-DS-6.7B |
| ise-uiuc/Magicoder-S-CL-7B | MagiCoderS-CL-7B | 01/01/2023 | Magicoder-S-CL-7B |
| bigcode/starcoder2-3b | StarCoder2-3b | 01/01/2023 | starcoder2-3b |
| bigcode/starcoder2-7b | StarCoder2-7b | 01/01/2023 | starcoder2-7b |
| bigcode/starcoder2-15b | StarCoder2-15b | 01/01/2023 | starcoder2-15b |

| codellama/CodeLlama-70b-hf | CodeLlama-70b-Base | 01/01/2023 | CodeLlama-70b-hf |
|---|---|---|---|
| codellama/CodeLlama-34b-hf | CodeLlama-34b-Base | 01/01/2023 | CodeLlama-34b-hf |
| codellama/CodeLlama-13b-hf | CodeLlama-13b-Base | 01/01/2023 | CodeLlama-13b-hf |
| codellama/CodeLlama-7b-hf | CodeLlama-7b-Base | 01/01/2023 | CodeLlama-7b-hf |
| deepseek-ai/deepseek-coder-33b-base | DSCoder-33b-Base | 08/30/2023 | deepseek-coder-33b-base |
| deepseek-ai/deepseek-coder-6.7b-base | DSCoder-6.7b-Base | 08/30/2023 | deepseek-coder-6.7b-base |
| deepseek-ai/deepseek-coder-1.3b-base | DSCoder-1.3b-Base | 08/30/2023 | deepseek-coder-1.3b-base |
| google/codegemma-7b | CodeGemma-7b-Base | 01/01/2023 | codegemma-7b |
| google/codegemma-2b | CodeGemma-2b-Base | 01/01/2023 | codegemma-2b |
| google/gemma-7b | Gemma-7b-Base | 01/01/2023 | gemma-7b |
| google/gemma-2b | Gemma-2b-Base | 01/01/2023 | gemma-2b |
| meta-llama/Meta-Llama-3-70B | LLama3-70b-Base | 01/01/2023 | Meta-Llama-3-70B |
| meta-llama/Meta-Llama-3-8B | LLama3-8b-Base | 01/01/2023 | Meta-Llama-3-8B |
| mistral-large-latest | Mistral-Large | 01/01/2023 | mistral-large-latest |
| open-mixtral-8x22b | Mixtral-8x22B-Ins | 01/01/2023 | open-mixtral-8x22b |
| open-mixtral-8x7b | Mixtral-8x7B-Ins | 01/01/2023 | open-mixtral-8x7b |
| m-a-p/OpenCodeInterpreter-DS-33B | OC-DS-33B | 08/30/2023 | OpenCodeInterpreter-DS-33B |
| m-a-p/OpenCodeInterpreter-DS-6.7B | OC-DS-6.7B | 08/30/2023 | OpenCodeInterpreter-DS-6.7B |
| m-a-p/OpenCodeInterpreter-DS-1.3B | OC-DS-1.3B | 08/30/2023 | OpenCodeInterpreter-DS-1.3B |
| command-r | Command-R | 01/01/2023 | command-r |
| command-r+ | Command-R+ | 01/01/2023 | command-r+ |

Table 3: Language Models Overview

We use variety of GPUs based on availability for running local models (A6000, L4, A100).

## D.2 CODE GENERATION

Below we provide the prompt format (with appropriate variants adding special tokens accommodating each instruct-tuned model) used for this scenario.

## D.3 SELF REPAIR

Below we provide the prompt format (with appropriate variants adding special tokens accommodating each instruct-tuned model) used for this scenario.

## D.4 CODE EXECUTION

Below we provide the prompts for code execution with and without CoT. The prompts are modified versions of those from (Gu et al., 2024) to fit the format of the samples in our benchmark.

## D.5 TEST OUTPUT PREDICTION

Below we provide the prompt format (with appropriate variants adding special tokens accommodating each instruct-tuned model) used for this scenario.

```
You are an expert Python programmer. You will be given a question (
    ↪ problem specification) and will generate a correct Python
    ↪ program that matches the specification and passes all tests. You
    ↪  will NOT return anything except for the program

### Question:\n{question.question_content}

{ if question.starter_code }
 ### Format: {PromptConstants.FORMATTING_MESSAGE}

```python
{question.starter_code}
```
{ else }
### Format: {PromptConstants.FORMATTING_WITHOUT_STARTER_MESSAGE}

```python
# YOUR CODE HERE
```
{ endif }

### Answer: (use the provided format with backticks)
```

**Code Generation Prompt**

```
{if check_result.result_status is "Wrong Answer"}
The above code is incorrect and does not pass the testcase.
Input: {wrong_testcase_input}
Output: {wrong_testcase_output}
Expected: {wrong_testcase_expected}

{elif check_result.result_status is "Time Limit Exceeded"}
The above code is incorrect and exceeds the time limit.
Input: {wrong_testcase_input}

{elif check_result.result_status is "Runtime Error"}
The above code is incorrect and has a runtime error.
Input: {wrong_testcase_input}
Error Message: {wrong_testcase_error_message}

{endif}
```

**Self Repair Error Feedback Pseudocode**

```
You are a helpful programming assistant and an expert Python programmer
    ↪ . You are helping a user write a program to solve a problem. The
    ↪  user has written some code, but it has some errors and is not
    ↪ passing the tests. You will help the user by first giving a
    ↪ concise (at most 2-3 sentences) textual explanation of what is
    ↪ wrong with the code. After you have pointed out what is wrong
    ↪ with the code, you will then generate a fixed version of the
    ↪ program. You must put the entire fixed program within code
    ↪ delimiters only once.

### Question:\n{question.question_content}

### Answer: ```python
{code.code_to_be_corrected}
```

### Format: {PromptConstants.FORMATTING_CHECK_ERROR_MESSAGE}

### Answer: (use the provided format with backticks)
```

**Self-Repair Prompt**

```
You are given a Python function and an assertion containing an input to
    ↪  the function. Complete the assertion with a literal (no
    ↪ unsimplified expressions, no function calls) containing the
    ↪ output when executing the provided code on the given input, even
    ↪  if the function is incorrect or incomplete. Do NOT output any
    ↪ extra information. Provide the full assertion with the correct
    ↪ output in [ANSWER] and [/ANSWER] tags, following the examples.

[PYTHON]
def repeatNumber(number : int) -> int:
    return number
assert repeatNumber(number = 17) == ??
[/PYTHON]
[ANSWER]
assert repeatNumber(number = 17) == 17
[/ANSWER]

[PYTHON]
def addCharacterA(string : str) -> str:
    return string + "a"
assert addCharacterA(string = "x9j") == ??
[/PYTHON]
[ANSWER]
assert addCharacterA(string = "x9j") == "x9ja"
[/ANSWER]

[PYTHON]
{code}
assert {input} == ??
[/PYTHON]
[ANSWER]
```

**Code Execution Prompt**

```
You are given a Python function and an assertion containing an input to
    ↪  the function. Complete the assertion with a literal (no
    ↪ unsimplified expressions, no function calls) containing the
    ↪ output when executing the provided code on the given input, even
    ↪  if the function is incorrect or incomplete. Do NOT output any
    ↪ extra information. Execute the program step by step before
    ↪ arriving at an answer, and provide the full assertion with the
    ↪ correct output in [ANSWER] and [/ANSWER] tags, following the
    ↪ examples.

[PYTHON]
def performOperation(s):
    s = s + s
    return "b" + s + "a"
assert performOperation(s = "hi") == ??
[/PYTHON]
[THOUGHT]
Let's execute the code step by step:

1. The function performOperation is defined, which takes a single
    ↪ argument s.
2. The function is called with the argument "hi", so within the
    ↪ function, s is initially "hi".
3. Inside the function, s is concatenated with itself, so s becomes "
    ↪ hihi".
4. The function then returns a new string that starts with "b",
    ↪ followed by the value of s (which is now "hihi"), and ends with
    ↪ "a".
5. The return value of the function is therefore "bhihia".
[/THOUGHT]
[ANSWER]
assert performOperation(s = "hi") == "bhihia"
[/ANSWER]

[PYTHON]
{code}
assert {input} == ??
[/PYTHON]
[THOUGHT]
```

Code Execution Prompt with CoT

```
### Instruction: You are a helpful programming assistant and an expert
    ↪ Python programmer. You are helping a user to write a test case
    ↪ to help to check the correctness of the function. The user has
    ↪ written a input for the testcase. You will calculate the output
    ↪ of the testcase and write the whole assertion statement in the
    ↪ markdown code block with the correct output.

Problem:
{problem_statement}

Function:
```
{function_signature}
```
Please complete the following test case:

```
assert {function_name}({testcase_input}) == # TODO
```
### Response:
```

**Test Output Prediction Prompt**

# E    RESULTS

## E.1    CONTAMINATION

Figure 11 demonstrates contamination in DEEPSEEK in self repair and test output prediction scenarios.

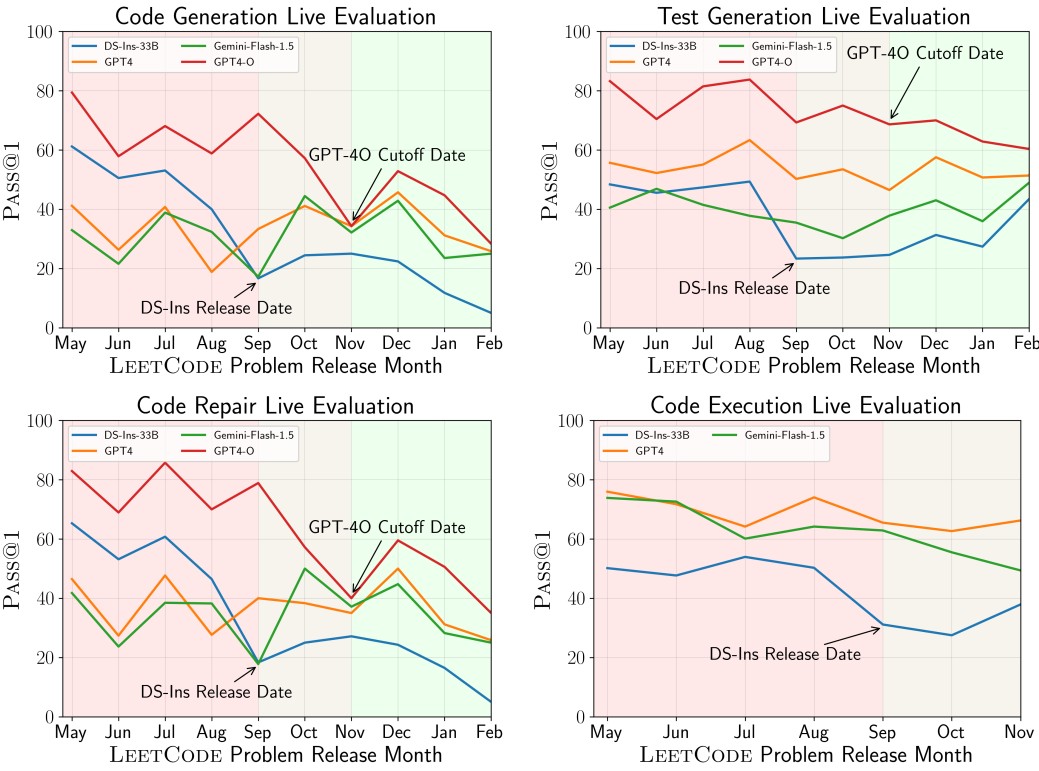

Figure 11: Contamination in DS-B models across self-repair and code execution (without COT) scenarios over time. Note that code execution currently runs between May and November

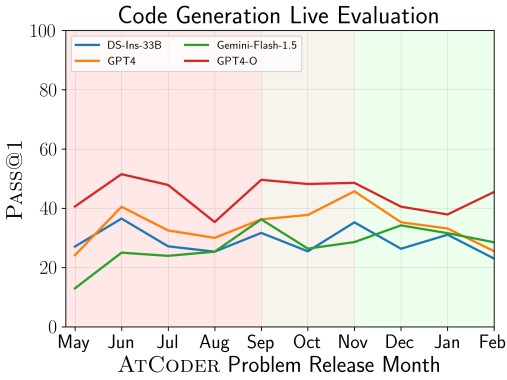

Figure 12: Performance on problems released over different months for ATCODER

## E.2    ALL RESULTS

Below we provide the tables comprising of results across different LIVECODEBENCH scenarios.

**Comparing Closed Models.** We evaluate a range of closed (API access) models ranging from different model families like GPTs, CLAUDES, GEMINI, and MISTRAL. We find the GPT-4-TURBO and

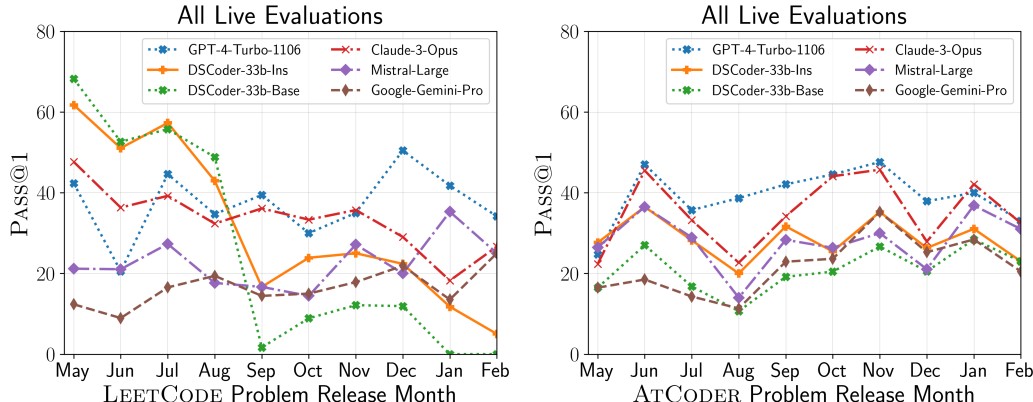

Figure 13: Live evaluation over time for various models on code generation scenario in LIVE-CODEBENCH. We consider many recently released models and do not find significant performance variations across months except for DS-B models.

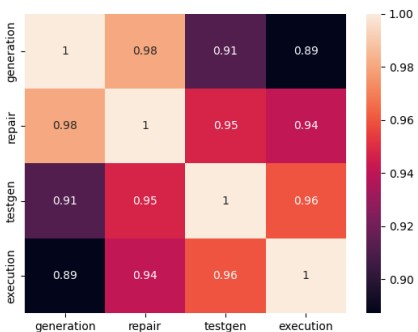

Figure 14: Correlations across different scenarios studied in LIVECODEBENCH

CLAUDE-3-OPUS rank at the top across all scenarios followed by MISTRAL-L and CLAUDE-3-SONNET models. Finally, GEMINI-PRO and GPT-3.5-TURBO lie on the lower end of the models. The relative differences between the models vary across the scenarios. For example, GPT-4-TURBO demonstrates remarkable improvement from self-repair (24.5% to 36.9% on the LCB-Medium problems) while GEMINI-PRO only improves from 8.5% to 9.4%. Similarly, as identified above, CLAUDE-3-OPUS and MISTRAL-L perform considerably better on test output prediction and code execution scenarios.

### E.3  C++ RESULTS

We evaluate model performances with C++ in Table 8. As we see, model performances roughly align across the two languages.

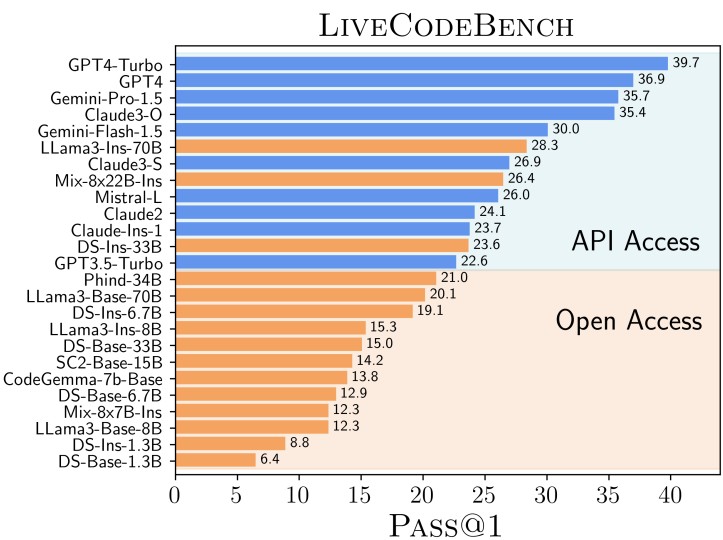

Figure 15: Comparison of open access and (closed) API access models on LiveCodeBench-Easy code generation scenario. We find that closed-access models consistently outperform the open models with only strong instruction-tuned variants of $> 30$B models (specifically L3-Ins-70B, Mixtral and DS-Ins-33B models) crossing the performance gap.

| Model Name | Easy | Medium | Hard | Total |
|---|---|---|---|---|
| Claude-2 | 61.80 | 4.90 | 0.20 | 22.30 |
| Claude-3-Haiku | 63.00 | 4.30 | 1.10 | 22.80 |
| Claude-3-Opus | 78.80 | 16.30 | 3.20 | 32.80 |
| Claude-3-Sonnet | 67.60 | 6.20 | 1.10 | 25.00 |
| Claude-Instant-1 | 60.70 | 4.30 | 1.10 | 22.10 |
| CodeGemma-2b-Base | 18.30 | 0.40 | 0.00 | 6.30 |
| CodeGemma-7b-Base | 35.70 | 2.60 | 0.10 | 12.80 |
| CodeLlama-13b-Base | 24.60 | 0.90 | 0.00 | 8.50 |
| CodeLlama-13b-Ins | 36.60 | 2.40 | 0.00 | 13.00 |
| CodeLlama-34b-Base | 32.20 | 1.80 | 0.10 | 11.40 |
| CodeLlama-34b-Ins | 33.70 | 2.40 | 1.10 | 12.40 |
| CodeLlama-70b-Base | 15.80 | 1.20 | 0.00 | 5.70 |
| CodeLlama-70b-Ins | 7.80 | 0.60 | 0.00 | 2.80 |
| CodeLlama-7b-Base | 19.00 | 0.40 | 0.00 | 6.50 |
| CodeLlama-7b-Ins | 28.60 | 2.50 | 0.00 | 10.40 |
| CodeQwen15-7B | 40.40 | 4.80 | 0.00 | 15.10 |
| CodeQwen15-7B-Chat | 39.20 | 13.10 | 0.50 | 17.60 |
| Codestral-Latest | 69.00 | 18.70 | 0.90 | 29.50 |
| Command-R | 39.00 | 3.60 | 0.00 | 14.20 |
| Command-R+ | 56.60 | 6.80 | 0.00 | 21.10 |
| DSCoder-1.3b-Base | 17.30 | 0.70 | 0.00 | 6.00 |
| DSCoder-1.3b-Ins | 22.90 | 1.50 | 0.00 | 8.10 |
| DSCoder-33b-Base | 39.40 | 2.30 | 0.00 | 13.90 |
| DSCoder-33b-Ins | 55.60 | 9.00 | 0.70 | 21.80 |
| DSCoder-6.7b-Base | 34.30 | 1.40 | 0.10 | 11.90 |
| DSCoder-6.7b-Ins | 46.40 | 5.80 | 0.70 | 17.60 |
| GPT-3.5-Turbo-0125 | 56.80 | 10.80 | 0.10 | 22.60 |
| GPT-3.5-Turbo-0301 | 53.40 | 8.80 | 0.20 | 20.80 |
| GPT-4-0613 | 78.40 | 21.20 | 2.30 | 33.90 |
| GPT-4-Turbo-1106 | 84.40 | 24.00 | 0.50 | 36.30 |
| GPT-4-Turbo-2024-04-09 | 85.30 | 33.00 | 5.10 | 41.10 |
| GPT-4O-2024-05-13 | 88.30 | 33.20 | 4.20 | 41.90 |
| Gemini-Flash-1.5-May | 68.10 | 12.60 | 2.70 | 27.80 |
| Gemini-Pro-1.5-April (n=1) | 56.50 | 14.30 | 3.60 | 24.80 |
| Gemini-Pro-1.5-May | 76.00 | 19.40 | 3.50 | 33.00 |
| Gemma-2b-Base | 6.10 | 0.00 | 0.00 | 2.00 |
| Gemma-7b-Base | 27.00 | 0.90 | 0.00 | 9.30 |
| LLama3-70b-Base | 52.20 | 3.20 | 0.60 | 18.60 |
| LLama3-70b-Ins | 60.70 | 15.80 | 1.40 | 26.00 |
| LLama3-8b-Base | 32.90 | 1.50 | 0.00 | 11.50 |
| LLama3-8b-Ins | 38.60 | 3.50 | 0.50 | 14.20 |
| MagiCoderS-CL-7B | 32.80 | 2.40 | 0.00 | 11.70 |
| MagiCoderS-DS-6.7B | 49.20 | 7.50 | 0.00 | 18.90 |
| Mistral-Large | 60.20 | 10.90 | 0.90 | 24.00 |
| Mixtral-8x22B-Ins | 59.80 | 12.70 | 0.00 | 24.20 |
| Mixtral-8x7B-Ins | 31.60 | 2.60 | 0.00 | 11.40 |
| OC-DS-1.3B | 11.30 | 0.10 | 0.00 | 3.80 |
| OC-DS-33B | 53.90 | 5.10 | 0.00 | 19.70 |
| OC-DS-6.7B | 46.30 | 4.50 | 0.00 | 16.90 |
| Phind-34B-V2 | 53.40 | 4.70 | 0.10 | 19.40 |
| StarCoder2-15b | 37.30 | 2.20 | 0.00 | 13.20 |
| StarCoder2-3b | 28.20 | 0.70 | 0.00 | 9.60 |
| StarCoder2-7b | 29.90 | 1.20 | 0.00 | 10.40 |

Table 4: Code Generation Performances

| Model Name | Easy | Medium | Hard | Total |
|---|---|---|---|---|
| Claude-2 | 66.20 | 10.30 | 0.40 | 25.60 |
| Claude-3-Haiku | 66.50 | 8.70 | 2.50 | 25.90 |
| Claude-3-Opus | 83.10 | 23.70 | 6.70 | 37.80 |
| Claude-3-Sonnet | 72.60 | 11.80 | 2.20 | 28.90 |
| Claude-Instant-1 | 64.40 | 7.10 | 2.20 | 24.60 |
| CodeLlama-13b-Ins | 43.10 | 3.00 | 0.00 | 15.30 |
| CodeLlama-34b-Ins | 31.50 | 3.50 | 1.80 | 12.30 |
| CodeLlama-7b-Ins | 31.90 | 3.10 | 1.50 | 12.10 |
| Codestral-Latest | 72.50 | 25.90 | 3.30 | 33.90 |
| DSCoder-1.3b-Ins | 29.50 | 2.10 | 0.00 | 10.60 |
| DSCoder-33b-Ins | 60.70 | 8.10 | 1.50 | 23.40 |
| DSCoder-6.7b-Ins | 49.90 | 5.70 | 1.10 | 18.90 |
| GPT-3.5-Turbo-0125 | 59.30 | 11.90 | 0.50 | 23.90 |
| GPT-3.5-Turbo-0301 | 58.40 | 11.60 | 0.70 | 23.60 |
| GPT-4-0613 | 79.30 | 25.00 | 2.40 | 35.60 |
| GPT-4-Turbo-1106 | 86.90 | 36.90 | 4.00 | 42.60 |
| GPT-4-Turbo-2024-04-09 | 88.70 | 39.70 | 8.40 | 45.60 |
| GPT-4O-2024-05-13 | 92.60 | 46.40 | 8.20 | 49.10 |
| Gemini-Flash-1.5-May | 73.40 | 16.40 | 4.40 | 31.40 |
| Gemini-Pro | 53.80 | 9.40 | 0.20 | 21.10 |
| Gemini-Pro-1.5-April (n=1) | 71.80 | 19.40 | 5.50 | 32.20 |
| Gemini-Pro-1.5-May | 84.80 | 30.10 | 7.30 | 40.70 |
| LLama3-70b-Ins | 69.60 | 19.00 | 1.80 | 30.10 |
| LLama3-8b-Ins | 47.10 | 6.10 | 0.00 | 17.70 |
| MagiCoderS-CL-7B | 36.50 | 3.10 | 0.00 | 13.20 |
| MagiCoderS-DS-6.7B | 50.60 | 8.60 | 0.00 | 19.70 |
| Mistral-Large | 71.20 | 15.60 | 3.60 | 30.10 |
| OC-DS-1.3B | 20.00 | 0.40 | 0.00 | 6.80 |
| OC-DS-33B | 58.90 | 7.20 | 1.30 | 22.50 |
| OC-DS-6.7B | 50.90 | 6.30 | 0.20 | 19.10 |
| Phind-34B-V2 | 62.00 | 6.50 | 0.90 | 23.10 |

Table 5: Self Repair Performances

| Model Name | Pass@1 |
| --- | --- |
| Claude-2 | 32.70 |
| Claude-3-Haiku | 32.90 |
| Claude-3-Opus | 58.70 |
| Claude-3-Sonnet | 34.10 |
| Claude-Instant-1 | 25.40 |
| CodeLlama-13b-Ins | 24.40 |
| CodeLlama-34b-Ins | 23.00 |
| CodeLlama-70b-Ins | 16.10 |
| CodeLlama-7b-Ins | 15.30 |
| Codestral-Latest | 41.80 |
| DSCoder-1.3b-Ins | 12.50 |
| DSCoder-33b-Ins | 28.30 |
| DSCoder-6.7b-Ins | 26.50 |
| GPT-3.5-Turbo-0125 | 35.40 |
| GPT-3.5-Turbo-0301 | 32.50 |
| GPT-4-0613 | 52.90 |
| GPT-4-Turbo-1106 | 55.70 |
| GPT-4-Turbo-2024-04-09 | 66.10 |
| GPT-4O-2024-05-13 | 68.90 |
| Gemini-Flash-1.5-May | 38.10 |
| Gemini-Pro | 29.50 |
| Gemini-Pro-1.5-April (n=1) | 49.60 |
| Gemini-Pro-1.5-May | 44.80 |
| LLama3-70b-Ins | 41.40 |
| LLama3-8b-Ins | 24.40 |
| MagiCoderS-CL-7B | 21.30 |
| MagiCoderS-DS-6.7B | 27.10 |
| Mistral-Large | 46.50 |
| Mixtral-8x22B-Ins | 44.70 |
| Mixtral-8x7B-Ins | 31.80 |
| OC-DS-1.3B | 7.80 |
| OC-DS-33B | 11.30 |
| OC-DS-6.7B | 18.30 |
| Phind-34B-V2 | 27.20 |

Table 6: Test Output Prediction Performances

| Model Name | Pass@1 | Pass@1 (COT) |
|---|---|---|
| Claude-2 | 31.50 | 43.80 |
| Claude-3-Haiku | 0.70 | 28.30 |
| Claude-3-Opus | 36.50 | 80.10 |
| Claude-3-Sonnet | 29.30 | 39.40 |
| Claude-Instant-1 | 20.00 | 34.80 |
| Cllama-13b-Ins | 23.50 | 14.10 |
| Cllama-34b-Ins | 28.90 | 24.50 |
| Cllama-7b-Ins | 20.60 | 14.20 |
| CodeLlama-70b-Ins | 31.20 | -1.00 |
| Codestral-Latest | 37.90 | 41.80 |
| DSCoder-1.3b-Base | 19.00 | 13.40 |
| DSCoder-1.3b-Ins | 18.10 | 17.00 |
| DSCoder-33b-Base | 29.90 | 29.10 |
| DSCoder-33b-Ins | 26.60 | 31.70 |
| DSCoder-6.7b-Base | 23.50 | 25.10 |
| DSCoder-6.7b-Ins | 23.10 | 23.80 |
| GPT-3.5-Turbo-0301 | 33.90 | 34.80 |
| GPT-4-0613 | 44.30 | 64.80 |
| GPT-4-Turbo-1106 | 40.50 | 83.60 |
| GPT-4-Turbo-2024-04-09 | 45.90 | 83.80 |
| GPT-4O-2024-05-13 | 39.10 | 91.00 |
| Gemini-Flash-1.5-May | 21.40 | 57.10 |
| Gemini-Pro | 27.70 | 37.40 |
| Gemini-Pro-1.5 (April) (n=1) | 30.30 | 64.40 |
| Gemini-Pro-1.5-May | 42.10 | 72.10 |
| LLama3-70b-Ins | 29.60 | 55.50 |
| LLama3-8b-Ins | 18.40 | 29.40 |
| MagiCoderS-CL-7B | 21.20 | -1.00 |
| MagiCoderS-DS-6.7B | 27.20 | -1.00 |
| Mistral-Large | 36.60 | 54.40 |
| Phind-34B-V2 | 26.90 | -1.00 |
| StarCoder | 20.30 | -1.00 |
| WCoder-34B-V1 | 28.40 | -1.00 |

Table 7: Code Execution Performances

| Model | Pass@1 |
|---|---|
| GPT-4o | 49.8 |
| Claude-3.5-Sonnet | 45.9 |

Table 8: Results for models with C++ evaluations on the entire 713 LiveCodeBench problems

# F QUALITATIVE EXAMPLES

## F.1 CODE EXECUTION

We show 5 examples from the code execution task that GPT-4 (`gpt-4-1106-preview`) still struggles to execute, even with CoT.

```python
def countWays(nums: List[int]) -> int:
    nums.sort()
    n = len(nums)
    ans = 0
    for i in range(n + 1):
        if i and nums[i-1] >= i: continue
        if i < n and nums[i] <= i: continue
        ans += 1
    return ans
assert countWays(nums = [6, 0, 3, 3, 6, 7, 2, 7]) == 3
# GPT-4 + CoT Outputs: 1, 2, 4, 5
```
**Mistake 1**

```python
def minimumCoins(prices: List[int]) -> int:

    @cache
    def dfs(i, free_until):
        if i >= len(prices):
            return 0

        res = prices[i] + dfs(i + 1, min(len(prices) - 1, i + i + 1))

        if free_until >= i:
            res = min(res, dfs(i + 1, free_until))

        return res

    dfs.cache_clear()
    return dfs(0, -1)
assert minimumCoins(prices = [3, 1, 2]) == 4
# GPT-4 + CoT Outputs: 1, 3, 5, 6
```
**Mistake 2**

```python
def sortVowels(s: str) -> str:
    q = deque(sorted((ch for ch in s if vowel(ch))))
    res = []
    for ch in s:
        if vowel(ch):
            res.append(q.popleft())
        else:
            res.append(ch)
    return ''.join(res)
assert sortVowels(s = 'lEetcOde') == 'lEOtcede'
# GPT-4 + CoT Outputs: "leetecode", "lEetecOde", "leetcede", "leetcEde
    ↪ ", "leetcOde"
```
**Mistake 3**

```
def relocateMarbles(nums: List[int], moveFrom: List[int], moveTo: List[
    ↪ int]) -> List[int]:

    nums = sorted(list(set(nums)))
    dd = {}
    for item in nums:
        dd[item] = 1
    for a,b in zip(moveFrom, moveTo):
        del dd[a]
        dd[b] = 1
    ll = dd.keys()
    return sorted(ll)
assert relocateMarbles(nums = [1, 6, 7, 8], moveFrom = [1, 7, 2],
    ↪ moveTo = [2, 9, 5]) == [5, 6, 8, 9]
# GPT-4 + CoT Outputs: [2, 6, 8, 9], [2, 5, 6, 8, 9], KeyError
```

**Mistake 4**

```
def minimumSum(nums: List[int]) -> int:
    left, right, ans = [inf], [inf], inf
    for num in nums:
        left.append(min(left[-1], num))
    for num in nums[::-1]:
        right.append(min(right[-1], num))
    right.reverse()
    for i, num in enumerate(nums):
        if left[i] < num and right[i + 1] < num:
            ans = min(ans, num + left[i] + right[i + 1])
    return ans if ans < inf else -1
assert minimumSum(nums = [6, 5, 4, 3, 4, 5]) == -1
# GPT-4 + CoT Outputs: 10, 11, 12
```

**Mistake 5**

## G    LIMITATIONS

**Benchmark Size.** LIVECODEBENCH code generation scenario currently hosts over 500 instances from problems released since May 2023. To account for contamination in DEEPSEEK, we only perform evaluations on problems released after the model cutoff date. This leads to only 349 problems used in our final evaluations which might add noise due to problem set samples. We currently estimate $1 - 2\%$ performance variations in LCB code generation due to this issue (measured by bootstrapping 349 sized problem sets from the full dataset). Other scenarios, i.e. self-repair, code execution, and test output prediction comprise 238, 188, and 254 problems would have similar performance variations. We thus recommend exercising proper judgement when comparing models with small performance differences. Note that HUMANEVAL has 164 problems and would also struggle with similar issues.

This issue is also exacerbated for newer models, with more recent cutoff dates, as they might only have access to a smaller evaluation set. We propose two solutions addressing this issue as we evolve LIVECODEBENCH. First, we will use other competition platforms for problem collection, allowing larger number of recent problems to be added to the benchmark. In addition, we also hope supplement this with an unreleased private test set constructed specifically for model evaluation. These problems will use a similar flavor to current problems and will be used when models are submitted for evaluation to the LIVECODEBENCH platform. This would reduce the reliance on public accessible problems and provide a more robust evaluation of the models while providing community public access to similar problems, similar to strategies employed by popular platforms like KAGGLE.

**Focus on PYTHON.** LIVECODEBENCH currently only focuses on PYTHON which might not provide enough signal about model capabilities in other languages. However, since we collected problem statements and serialized tests, adding new programming languages would be straightforward once appropriate evaluation engines are used.

**Robustness to Prompts.** Recent works have identified huge performance variances that can be caused due to insufficient prompt. Here, we either do not tune prompts across models or make minor adjustments based on the system prompts and delimiter tokens. This can lead to performance variance in our results. Our findings and model comparison orders generalize across LIVECODEBENCH scenarios and mostly match the performance trends observed on HUMANEVAL making this a less prominient issue.

This issue can be particularly observed open models on the code execution scenario with COT prompting. Interestingly, often the open models perform even worse in comparsion to the direct code execution baseline. Note that we used same prompts for the closed models all of which show noticable improvement from COT. While the used prompts might be sub-optimal, this highlights how open-models perform worse against the closed models at performing chain-of-thought.

**Problem Domain.** Programming is a vast domain and occurs in various forms such as programming puzzles, competition programming, and real-world software development. Different domains might have individual requirements, constraints, challenges, and difficulty levels. LIVECODEBENCH currently focuses on competition problems sourced from three platforms. This might not be representative of the "most general" notion of LLM programming capabilities. Particularly, real-world usage of LLMs is drawn upon open-ended and unconstrained problems rasied by users. We therefore recommend using LIVECODEBENCH as a starting point for evaluating LLMs and further using domain-specific evaluations to measure and compare LLMs in specific settings as required.

