# OpenReview forum: "LiveCodeBench: Holistic and Contamination Free Evaluation of Large Language Models for Code"
_ICLR.cc/2025/Conference — ICLR 2025 Poster_

### Official Review · Reviewer_9FRx · 2024-10-24

**Soundness:** 3
**Presentation:** 3
**Contribution:** 3
**Rating:** 6
**Confidence:** 4

**Summary:**

This paper introduces LiveCodeBench, a new benchmark designed to evaluate LLMs for code-related tasks.
LiveCodeBench addresses some key limitations of previous benchmarks,
including issues like data contamination, overfitting, saturation, and limited application range.
Unlike existing benchmarks, LiveCodeBench assesses not only code generation but also self-repair, code execution, and test output prediction.
Alongside proposing the benchmark,
the authors perform a comprehensive evaluation and analysis of the contamination and overfitting problem in popular LLMs.
The paper also introduces a difficulty-guided problem curation strategy,
which enhances the evaluation's effectiveness by creating clearer margins between model performances,
allowing for a more meaningful comparison.
The evaluation covers 52 models, with sizes ranging from 1.3B to 70B,
providing a fairer and more reliable understanding of model performance in code-related tasks.

**Strengths:**

* LiveCodeBench has a live update mechanism that mitigates data contamination and enables continuous growth.
  This mechanism ensures that the benchmark remains useful for evaluating newer models,
  even those with a knowledge cut-off date beyond its release.
* With the timestamps on the tasks, the authors conduct a novel "live" evaluation of code generation to directly address data contamination issues.
  This approach offers deeper insights into the models' actual coding capabilities, focusing on genuine problem-solving rather than mere memorization.
* The difficulty guided problem curation is particularly effective in revealing performance differences between models of similar sizes, making it practical to use for meaningful evaluation.
* The paper employs a clustering method to identify models that overfit HumanEval.
  While overfitting to older benchmarks is a well-known issue,
  the authors' method offers empirical evidence and analysis that specifically highlights the models affected by this problem.

**Weaknesses:**

* Novelty: The benchmark's construction primarily aims to create a newer and harder version of existing benchmarks.
  However, a key issue with LLM benchmarks for code is that Olympiad in Informatics (OI) programs are not typically representative of real-world software engineering and programming languages.
  OI programs often have a distinct style and differ in their learned representations compared to code from real software projects.
  This has led to a shift in focus in this research area from OI competitive programming to open-source software projects, as seen in recent work like SWE-bench [1] and RepoBench [2].

[1]: Jimenez, Carlos E., et al. "Swe-bench: Can language models resolve real-world github issues?." ICLR 2024.

[2]: Liu, Tianyang, et al. "Repobench: Benchmarking repository-level code auto-completion systems." ICLR 2024.

**Questions:**

1. The tasks of *code execution* and *test case output prediction* appear to be quite similar.
   According to the paper, the distinction lies in their prompt:
   code execution is to predict the output based on inputs and functions in *programming language*,
   while test case output prediction is to predict the output based on inputs and function descriptions in *natural language*. Given the similarity between these two tasks, is there any observable correlation in their results?
2. Test case generation typically is designed as predicting the entire test case from a given function implementation [3,4],
   which is more practical in real-world software engineering and testing applications,
   such as in tools like Copilot.
   I'm curious why the authors choose a different approach to evaluating test generation in the form of $(I, f_{\text{NL}}) \mapsto O$,
   where the input and natural language function descriptions are used to predict the output.
3. In the **Problem Difficulty** paragraph of Section 3.1,
   the authors state that they collected problems of varying difficulty levels as labeled by competition platforms.
   However, how did the authors address potential rating bias across these platforms?
   Given that different competition platforms are designed for distinct user groups and, as the authors note, CodeForces problems are generally harder than those on other platforms,
   there is likely a selection bias in difficulty ratings across different platforms.

[3]: Nie, Pengyu, et al. "Learning deep semantics for test completion." ICSE 2023.

[4]: Rao, Nikitha, et al. "CAT-LM training language models on aligned code and tests." ASE 2023.

---

> ### Author Response · Authors · 2024-11-23
>
> > Novelty: The benchmark's construction primarily aims to create a newer and harder version of existing benchmarks. However, a key issue with LLM benchmarks for code is that Olympiad in Informatics (OI) programs are not typically representative of real-world software engineering and programming languages.
>
> **Contributions**. Our work has two-fold contributions.
>
> - First, we carefully curated diverse problems with necessary difficulty distribution and tests. We used these problems to construct evaluation scenarios that allow measuring programming agent capabilities in isolation.
> - We rigorously analyze model performances on our evaluation set and provide empirical findings on contamination, overfitting in HumanEval, comparison across tasks, and various model comparisons. Notably, in contrast with prior work, we identify over-fitting from post-training datasets as opposed to previously studied pre-training corporas.
>
> **Comparison to software engineering benchmarks**. We believe that competition and interview style problems provide a complementary and challenging evaluation set to software engineering benchmarks.
>
> - **Different but complementary research investigations.** While primary research investigations on SWEBench have focused on designing optimal agent computer interfaces [1], research in competition programming problems is attempting to train models with such behaviours.
>
> - **Availability of High-Quality Problems** APPS, CodeContests, and TACO provide good quality and easy-to-use training datasets for competition problems with execution environments and test cases. Achieving such environments is possible but considerably more challenging for software engineering setup. Such easy-to-use training environments are beneficial for researchers and have allowed the exploration of various reinforcement learning-based solutions.
>
> Given such community interest, we believe LiveCodeBench provides a high-quality benchmark for evaluation competition programming capabilities of language models.
>
> > The tasks of code execution and test case output prediction appear to be quite similar.
>
> Yes, we plot the correlation between the tasks in Figure 13. As the reviewer observes, we find that code generation is more correlated with self-repair and test output prediction is more correlated with code execution scenarios.
>
> At the same time, we want to highlight the two scenarios that measure different skills. In particular, the natural language problem description does not provide algorithmic details for the solution. Thus, the model needs to jointly come up with a satisfying solution and “simulate” it while predicting tests.
>
> > Test case generation typically is designed as predicting the entire test case from a given function implementation [3,4],
>
> Thank you for the question! The test output prediction scenario is designed to evaluate the capabilities of code LLMs to create tests to filter *their* own incorrect solutions. Various “agentic” solutions like  CodeT and AlphaCodium have incorporated test generation and demonstrated its advantages. We design the test output prediction task to evaluate this capability in isolation.
>
> 1. Disentangling test inputs for test generation. As the reviewer mentioned, test generation comprises predicting both test inputs and corresponding outputs. Evaluating both aspects jointly is challenging and prior works have resorted to evaluating both coverage and test validity. Even then, since different test generation systems can generate independent inputs, the test validity scores may not be directly comparable. We believe that normalizing the test inputs allows for comparing test validity more rigorously – predicting test outputs on the same inputs.
>
> 2. Using more Inference time-compute to predict test outputs. Competition programming problems often involve challenging algorithms and math-heavy puzzle questions. Constructing test outputs for such problems is also non-trivial, and very close to MATH problems. Thus, as observed in MATH problems, we find that allowing the model to use more inference time using strategies like chain-of-thought, self-consistency considerably improves the test validity accuracies. For example, GPT-4O-Mini test output prediction accuracy is 61% which drops to 46% without chain-of-thought.

---

> > ### Author Response · Authors · 2024-11-23
> >
> > > In the Problem Difficulty paragraph of Section 3.1, how did the authors address potential rating bias across these platforms?
> >
> > Thank you for the question! We developed our difficulty thresholds by estimating model performances in different rating buckets. From our initial experiments and prior works that have attempted to use LeetCode for evaluations, we find that Easy, Medium, and Hard transfer well to LLM performance as well. Moreover, performance on LeetCode easy problems aligns well with HumanEval thus providing balanced problem difficulty such that current weaker models achieve non-zero performance.
> >
> > For Atcoder, we develop our problem rating brackets such that it similarly aligns with HumanEval and LeetCode problem difficulties.
> >
> > Performances for GPT-4-Turbo on Atocder and LeetCode problems
> >
> > | Platform | Overall Pass@1 | Easy Pass@1 | Medium Pass@1 | Hard Pass@1 |
> > |----------|---------------|-------------|---------------|------------|
> > | Atcoder  | 38.6% | 84.4% | 20.2% | 1.3% |
> > | LeetCode | 39.9% | 77.8% | 27.1% | 3.7% |
> >
> >
> > Thus, we use HumanEval scores and *relative* problem difficulty ratings on the platforms to calibrate difficulties across platforms.

---

> > ### Comment · Reviewer_9FRx · 2024-11-25
> >
> > Thanks to the authors for addressing my questions and additional experiments on the relative difficulty concern. I will keep my scores the same.

---

### Official Review · Reviewer_SgU1 · 2024-11-03

**Soundness:** 3
**Presentation:** 3
**Contribution:** 3
**Rating:** 8
**Confidence:** 4

**Summary:**

This paper worked on the evaluation of LLMs for code, proposing a benchmark named LiveCodeBench. The benchmark will be continuously updated by automatically collecting problems from online code contest sites to overcome data contamination. It contains four code-related tasks: code generation, self-repair, code execution, and test output prediction. A large-scale evaluation was conducted on 50 LLMs, which revealed the widespread contamination among LLMs, as well as overfitting and saturation of traditional coding benchmarks like HumanEval.

**Strengths:**

1. **Benefiting the community**: The benchmark will be continuously updated and provide code ability evaluation with less contamination.
2. **Large scale evaluation**: The authors evaluated 50 LLMs with carefulness about contamination, providing a valuable reference for the code-related capabilities of each model.
3. **Having insights**: The experiments revealed the widespread contamination among LLMs, as well as overfitting and saturation of traditional coding benchmarks like HumanEval.

**Weaknesses:**

1. **Unproven data representative**: Besides contamination, we also need to know how many code problems we need and how representative they are, to conduct a comprehensive evaluation. However, they were not answered in this paper. Are there mechanisms to guarantee or prove these properties, in particular for new models?

2. **Unevaluated workflow reliability**: While the workflow of benchmark construction is completely automated, this workflow's reliability was not evaluated. For example, the accuracy of the HTML extractor.

3. **Unproven test case completeness**: Averaged 18 of the test case count basically is far fewer than test cases used inside the programming task websites; the completeness or sufficiency of these test cases was not analyzed in this paper.

4. **Biased filtering**: In the code competition scenario that this work focused on, questions with multiple correct outputs for a single input would contain many unique features or have different problem-type distributions. However, they were all removed in this work, which might lead to a bias. Further analysis of the impact of this filtering should be conducted.

5. **Multi-step track needed**: This work's designs of four tracks were motivated by AlphaCodium. Including AlphaCodium, state-of-the-art competition-level code generation works nowadays broadly applied multi-step workflow. However, besides one-step revision, multi-step code generation was not evaluated in this work.

**Questions:**

The contribution of this work to the community depends largely on your continued maintenance efforts. Do you have a long-term plan?

---

> ### Author Response · Authors · 2024-11-23
>
> > Unproven data representative: Besides contamination, we also need to know how many code problems we need and how representative they are, to conduct a comprehensive evaluation. However, they were not answered in this paper. Are there mechanisms to guarantee or prove these properties, in particular for new models?
>
> - **Large and Growing Dataset:** LiveCodeBench currently hosts **over 700 problems** collected primarily from LeetCode and AtCoder platforms. Our dataset is continually expanding as we automate the collection of new problems.
>
> - **Statistical Significance:** In Appendix G, we present a bootstrapping analysis to measure evaluation noise. We found that using a subset of **338 problems** yields a low variance of **1-2%** in model comparisons. Utilizing the full suite of 713 problems further reduces this variance, enhancing the reliability of our evaluations.
>
> - **Diverse and Real-World Problems:** Our benchmark includes a wide range of problem types, such as algorithms, data structures, mathematical computations, and real-world scenarios. This diversity ensures that models are evaluated on various aspects of programming skills. Additionally, since these problems are sourced from popular platforms hosting such problems. Given humans regularly participate in such contests, these problems reflect how models would perform in such a 'real-world' setting.
>
> > Unevaluated workflow reliability: While the workflow of benchmark construction is completely automated, this workflow's reliability was not evaluated. For example, the accuracy of the HTML extractor.
>
> We followed existing best practices while extracting problem statements from HTML pages. In particular, we extracted equations, normalized markdown, removed duplicated problems, and filtered problems with images. When constructing our extractors, we also used LLMs to provide feedback by using them to evaluate any mistakes in parsing however did not find it very reliable.
>
> We manually scanned through 30 random problems in our collected benchmark and compared them with the rendered HTML webpage problems and did not find any errors in our collected problems.
>
> > Unproven test case completeness: Averaged 18 of the test case count basically is far fewer than test cases used inside the programming task websites; the completeness or sufficiency of these test cases was not analyzed in this paper.
>
> Thank you for the question. First, note that while we only provide 18 tests, we collect large and complex tests, often with inputs sized larger than $1e5$ elements using our generator-based input generation approach. Thus even single instances of such inputs can evaluate multiple diverse program behaviors.
>
> Next, we observed that the number of tests presented a tradeoff in terms of benchmark usability and evaluation precision. In particular, initially, LiveCodeBench provided up to 100 tests per problem and suffered from considerably longer evaluation times (over 10 minutes for computing pass@1 for 400 problems.
>
> Based on user feedback, we studied pass@1 performance with different counts of the sampled number of tests and found 20 tests to hit a sweet spot in terms of evaluation reliability and providing fast evaluation signals. We have added this analysis to Appendix A.4 and Figure 9.
>
> > Multi-step track needed: This work's designs of four tracks were motivated by AlphaCodium. Including AlphaCodium, state-of-the-art competition-level code generation works nowadays broadly applied multi-step workflow. However, besides one-step revision, multi-step code generation was not evaluated in this work.
>
> Thank you for the question. Indeed, multi-step repair is employed in current works. However, performing multi-turn repairs can be expensive, and introduces various design choices (e.g. which solutions to repair as studied in [1], type of revision feedback [2]). In this work, we therefore evaluate the revision capabilities of models using a baseline single-step repair setup. While a thorough analysis of various design choices is outside the scope of our work, our benchmark and evaluation setup can be easily extended to performing multi-turn evaluations.
>
> [1] Code Repair with LLMs gives an Exploration-Exploitation Tradeoff
> [2] What Makes Large Language Models Reason in (Multi-Turn) Code Generation?
>
> > The contribution of this work to the community depends largely on your continued maintenance efforts. Do you have a long-term plan?
>
> LiveCodeBench problem curation pipeline is completely automated (scraping, test generation) and can be easily run to collect new problems. Currently, the pipeline is run every 1-2 months, based on the sufficiency of the problems to curate new problems by the lead author.

---

> > ### Comment · Reviewer_SgU1 · 2024-12-02
> > **Feedback**
> >
> > Thank you for the rebuttal. I still hold a positive attitude towards this paper.

---

### Official Review · Reviewer_MFUT · 2024-11-04

**Soundness:** 3
**Presentation:** 3
**Contribution:** 2
**Rating:** 5
**Confidence:** 3

**Summary:**

Prior LLM coding benchmarks suffer from contamination and saturation issues. This paper introduces LiveCodeBench, a new benchmark that collects new problems from public programming contests (LeetCode, AtCoder, and Codeforces). Besides code completion, the benchmark also tests the self-repair and output prediction abilities of LLMs. The paper provides a comprehensive analysis on over 600 problems and over 50 LLMs, demonstrating that time-segmented evaluations are useful for detecting contamination.

**Strengths:**

- The paper proposes a coding benchmark with live updates, ensuring fair model evaluation by testing only on new problems after each model’s cutoff date. The time-segmented analysis also reveals a significant drop in performance of some models after release date, indicating notable contamination issues.
- The paper is well-written and organized, with a thorough appendix including discussion on legal compliance and benchmark creation details.

**Weaknesses:**

- The novelty is limited. Using competitive programming problems for LLM evaluation has been well adopted. Additionally, the evaluation scenarios in this paper are also not new but borrowed or adapted from prior work (e.g., the test case output prediction can be seen as a chain-of-thought prompting solution of the code execution task). The main contribution of this paper is to scrape the problems with appropriate difficulty and analyze the results in a time-segmented way.
- The relatively small number of problems (~40 LeetCode problems every two months) raises concern about the reliability of the results. I wonder if it’s possible to estimate the variance of the pass@1 performance and the statistical significance of the comparisons.

**Questions:**

It seems that the drop in performance occurs only for the LeetCode problems and is smooth for other platforms like AtCoder. Do you have any explanation for this interesting phenomenon?

---

> ### Author Response · Authors · 2024-11-23
>
> > Using competitive programming problems for LLM evaluation has been well adopted. Additionally, the evaluation scenarios in this paper are also not new but borrowed or adapted from prior work
>
> Our work has two-fold contributions.
> * As highlighted by the reviewer, first we carefully curated diverse but balanced difficulty problems. We used these problems to construct evaluation scenarios that allow measuring programming agent capabilities in isolation.
> * We rigorously analyze model performances on our evaluation set and provide empirical findings on contamination, overfitting in HumanEval, comparison across tasks, and various model comparisons. Notably, in contrast with prior work, we identify over-fitting from post-training datasets as opposed to previously studied pre-training corporas.
>
> > The relatively small number of problems (~40 LeetCode problems every two months) raises concern about the reliability of the results. I wonder if it’s possible to estimate the variance of the pass@1 performance and the statistical significance of the comparisons.
>
> Thank you for raising the important concern regarding reliability of our evaluations. We agree that using a sufficiently large and representative sample is crucial for statistical validity. In our study:
>
> - **Extensive Dataset Collection:** While Figure 1 illustrates performance over a two-month window using approximately 40 LeetCode problems to highlight contamination patterns, our overall evaluation is based on a much larger dataset. Specifically, since May 2023, we have collected a total of **713 problems** across multiple platforms, with **387 problems** gathered since the beginning of this year.
>
> - **Variance and Statistical Significance Estimation:** In Appendix G, we perform a detailed bootstrapped analysis to estimate the variance of the pass@1 performance. Our findings indicate that with **338 problems**, the error margins when comparing models are within **1-2%**. This margin decreases further with the full benchmark, enhancing the statistical significance of our comparisons.
>
> Finally, note that Figure 1 uses 40 LeetCode problems just to signify contamination. In general, LiveCodeBench evaluations average model performances on problems collected over multiple months from different platforms for reliable evaluations.
>
> > It seems that the drop in performance occurs only for the LeetCode problems and is smooth for other platforms like AtCoder. Do you have any explanation for this interesting phenomenon?
>
> Thank you for highlighting this intriguing observation. While we do not have direct access to the models' training datasets, we can offer some plausible explanations based on available information:
>
> - **Relative Popularity and Data Availability of LeetCode:** LeetCode is a widely used platform globally, particularly for technical interview preparation, with over 500,000 registered users. Consequently, there is an abundance of LeetCode problem solutions publicly available on platforms like GitHub. This widespread availability increases the likelihood that LeetCode problems and their solutions are included in the training data of LLMs, leading to potential overfitting or data contamination.
>
> - **Limited Exposure to AtCoder Problems:** In contrast, AtCoder is less prevalent internationally and is primarily utilized by a niche community of competitive programmers. As a result, AtCoder problems are less likely to be included in publicly available datasets, reducing the chance of contamination and overfitting.
>
> - **Training Data Selection Practices:** Previous research [1] indicates that early Code LLMs may have prioritized data from popular platforms like LeetCode during supervised fine-tuning. This selection bias could contribute to the observed performance discrepancies.
>
> [1] Evaluating Large Language Models Trained on Code

---

> > ### Comment · Reviewer_MFUT · 2024-11-26
> >
> > Thank you to the authors for their reply. I prefer to keep my current scores, as the responses did not fully address my concerns.
> >
> > As noted by the authors, the contamination analysis (Figure 1) represents the most interesting and core contribution of the paper, and that is why I am asking about the statistical validity at this particular scale of problems.

---

> > > ### Author Response · Authors · 2024-12-03
> > >
> > > Thank you for the follow-up.
> > >
> > > Our evidence for contamination follows from the noticable drops in performance observed in model performances over time windows before and after cutoff dates. We first compute the standard deviation of the bi-monthly model performance to demonstrate the significance. The following table lists the standard and max performance deviation of various models on problems released between May 2023 and August 2024.
> > >
> > > | Model | Std. (Bi-monthly Perf) | Max Dev. (Bi-monthly Perf) |
> > > |-------|-------------------------|----------------------------|
> > > | Gemini-Flash-1.5-May | 4.2 | 12.8 |
> > > | GPT-4-0613 | 5.7 | 14.9 |
> > > | Claude-3.5-Sonnet | 22.0 | 72.7 |
> > > | DS-Ins-33B | 15.7 | 49.2 |
> > > | GPT-4O-2024-05-13 | 13.5 | 43.6 |
> > >
> > >
> > >
> > > For GPT-4 and Gemini-Flash-2 with cutoff dates before May 2023, the standard deviation in bi-monthly model performances is less than 6. In contrast, for Claude-3.5-Sonnet, DS-33B, and GPT-4O the standard deviation is greater than 13 (and up to 22). Noticeably,  Claude3.5-Sonnet achieved 100% performance on Leetcode problems between May and June (including hard problems) and dropped more than 70 points on problems released from July to August.
> > >
> > > Using our bootstrap analysis, we identify around 10 points of noise when comparing models on 40 problems. The performance differences observed before and after the cutoff exceed that by a considerable margin. Particularly, the following table depicts model performances across two time-windows:
> > >
> > >
> > > | Metric |  #problems | DS-33b | GPT-4 | Gemini-Flash | GPT-4O | Claude-3.5-Sonnet |
> > > |--------|-----------------|-----------------|------------|----------------------|-------------------|-------------------|
> > > | May 2023 - Aug 2023  | 79 | 51.4% | 32.7% | 33.7% | 66.1% | 84.9% |
> > > | May 2024 - Aug 2024 | 64 | 20.2% | 31.1% | 28.9% | 43.3% | 33.9% |
> > >
> > > DS-33B outperforms GPT-4 by more than 20 points in the (contaminated) time window but is **worse** by 10 points in the newer window. These performance comparisons are significant: p-values 0.04 and 3e-5 on the respective time windows (computed using a two-sided binomial test) and flip the model rankings.
> > >
> > > Similar findings hold for other contaminated models as well.

---

### Official Review · Reviewer_zPuZ · 2024-11-04

**Soundness:** 4
**Presentation:** 4
**Contribution:** 2
**Rating:** 6
**Confidence:** 5

**Summary:**

The paper introduces LiveCodeBench, a lively-updated dynamic benchmark that is designed to evaluate large language models (LLMs) for coding tasks with minimized data contamination risk.

Aside from being contamination-free, LiveCodeBench also seeks to improve the data quality, difficulty, and task diversity of the benchmark.

The paper also conducts thorough experiments exposing the significant data contamination and overfitting issues.

**Strengths:**

## Significance

The paper addresses one of the most important issues: data contamination in the existing Code LLM evaluation process. The benchmark is highly practical and should be known and tried by the Code LLM community to encourage more rigorous evaluation of models.


## Soundness

The paper demonstrates strong soundness through its detailed analyses of the models' performance over time, overfitting issues on HumanEval, and comparison of different types of models.


## Effectiveness

Dynamic benchmarking is indeed an effective solution to mitigate data contamination.

**Weaknesses:**

## Novelty

The paper presents two key contributions:

1. A comprehensive dynamic benchmark for coding takes

2. Analyses exposing the contamination issues in coding tasks

While these contributions are highly relevant and practical for the Code LLM community, it's important to note that dynamic benchmarking is a technique that has been utilized in various prior works [1,2,3,4] as early as 2021 [1]. Additionally, the issue of data contamination has been systematically examined in earlier studies [5,6], limiting the novel insights that the paper can offer to the broader ICRL community.

To enhance the paper's novelty, it may be beneficial to develop better and more efficient dynamic benchmarking techniques, or introduce new methods to alleviate more intricate contamination issues such as re-phrasing [6].




[1] [Dynaboard: An Evaluation-As-A-Service Platform for Holistic Next-Generation Benchmarking](https://arxiv.org/abs/2106.06052)

[2] [Competition-Level Problems are Effective LLM Evaluators](https://arxiv.org/abs/2312.02143)

[3] [LatestEval: Addressing Data Contamination in Language Model Evaluation through Dynamic and Time-Sensitive Test Construction](https://arxiv.org/abs/2312.12343)

[4] [NPHardEval: Dynamic Benchmark on Reasoning Ability of Large Language Models via Complexity Classes](https://arxiv.org/abs/2312.14890)

[5] [To the Cutoff... and Beyond? A Longitudinal Perspective on LLM Data Contamination](https://openreview.net/forum?id=m2NVG4Htxs)

[6] [Rethinking Benchmark and Contamination for Language Models with Rephrased Samples](https://arxiv.org/abs/2311.04850)

**Questions:**

## Discussion Questions

Have you considered addressing more intricate data contamination issues like re-phrasing within the scope of this work?

I believe that many code-specific techniques can be developed to detect re-phrasing contamination such as checking the equivalence or similarity of the test cases and canonical solutions.

I would be happy to raise my score if the authors could incorporate and address those intricate contamination issues in the revision.

---

> ### Author Response · Authors · 2024-11-23
>
> > While these contributions are highly relevant and practical for the Code LLM community, it's important to note that dynamic benchmarking is a technique that has been utilized in various prior works [1,2,3,4] as early as 2021 [1].
>
> Thank you for bringing these references to our attention. We have updated the manuscript to appropriately contextualize prior work on dynamic benchmarking. Our work extends dynamic benchmarking to the Code LLM setting (emphasized highly relevant and practical by the reviewer). Note that beyond code generation, we also study (or introduce) other scenarios like repair, test generation, and code execution -- allowing measuring programming agent capabilities in isolation.
>
> > Additionally, the issue of data contamination has been systematically examined in earlier studies [5,6], limiting the novel insights that the paper can offer to the broader ICRL community.
>
> We appreciate your observation regarding prior studies on data contamination in code LLMs [5,6]. Our work builds and extends findings by shifting the focus from contamination in pre-training corpora to overfitting issues arising from post-training datasets. Specifically, we demonstrate that fine-tuned models, such as DeepSeekCoder, while achieving state-of-the-art performance on benchmarks like HumanEval, show significantly reduced performance even on LiveCodeBench-Easy where most closed models perform similarly to HumanEval.
>
> Indeed, prior works mentioned by the reviewer have studied the contamination in code LLMs.
>
> However, these works primarily focused on detecting contamination from samples in pre-training corpora. We expand those findings and in contrast, emphasize over-fitting from post-training datasets. In particular, while various fine-tuned open models achieve SOTA performance on HumanEval (e.g. DeepSeekCoder) they performed considerably worse even on LiveCodeBench Easy problems.
> We further identified that this gap widens after fine-tuning – i.e. base models are well correlated on HumanEval and LiveCodeBench-Easy but start to overfit on HumanEval after post-training (see “Comparing Base Models” and “Role of Post Training” paragraphs in Section 5).
>
> Beyond contamination, we also provide insights on model performance trends, contamination, and programming agent scenarios through a rigorous analysis of over 50 models in Section 5.
>
> > introduce new methods to alleviate more intricate contamination issues such as re-phrasing [6]. Have you considered addressing more intricate data contamination issues like re-phrasing within the scope of this work?
>
> Thank you for the suggestion. Unfortunately, we operate under full black-box access to underlying models, i.e. we do not have access to training (pre-training or post-training) datasets, model weights, or even log-probs. Hence, we can only identify contamination by measuring model performances on different sets of problems (time-segmented evaluations studied in this work). Since rephrasing contamination detection would require access to the training corpora, it is not immediately clear how to apply it in our scenario. Please let us know if we misunderstood your point.

---

> > ### Comment · Reviewer_zPuZ · 2024-11-25
> >
> > Thank the authors for the detailed response. I tend to keep my original scores.
> >
> > A few clarifications regarding the rephrasing contamination:
> >
> > 1. Under the full black-box setting, we can still study and alleviate rephrasing contamination by assuming the models have been trained on all the public Internet data before its training cutoff.
> > 2. It is highly likely that a newly released leetcode problem is just a rephrased version of an old problem on some other competitive programming websites.
> > 3. It would be beneficial for LiveCodeBench to detect those **"pseudo-new"** problems and exclude them from the benchmark.
> >
> > I encourage the authors to explore along this line, which will significantly increase the novelty of the work.

---

### Meta-Review · Area_Chair_V622 · 2024-12-19

**Metareview:**

The paper provides a benchmark for evaluating coding capabilities of language models. The benchmark focuses on large range of capabilities going beyond code generation. It is already being widely used by the community which is a big marker for usefulness. However, I still request the authors to update the camera ready version of the paper based on comments from the reviewers.

**Additional Comments On Reviewer Discussion:**

Reviewers had some concerns about data representativeness and statistical validity (LeetCode) which were addressed by the authors. Reviewer SgU1 raised concerns about HTML extraction accuracy and test case completeness and authors addressed by explaining manual verification of 30 random problems found no extraction errors. Overall, all reviewers were positive about the paper.

---

### Decision · Program_Chairs · 2025-01-22

Accept (Poster)